



# Characterization of the UV radiometric calibration for the TROPOMI operational ozone profile retrieval algorithm

Serena Di Pede [1,2], Erwin Loots[1], Emiel van der Plas[1], Maarten Sneep[1], Edward van Amelrooy[1], Mirna van Hoek[1], Mark ter Linden[1], Antje Ludewig[1], Arno Keppens[3], and J. Pepijn Veefkind[1,2]

[1]R&D Satellite Observations, Royal Netherlands Meteorological Institute (KNMI), De Bilt, The Netherlands
[2]Delft University of Technology, Delft, The Netherlands
[3]Royal Belgian Institute for Space Aeronomy (BIRA-IASB), Uccle, Belgium

**Correspondence:** Serena Di Pede (serena.dipede@knmi.nl)

**Abstract.** The European Space Agency (ESA) Sentinel-5 Precursor (S5P) is a low Earth orbit polar satellite carrying the single payload instrument TROPOspheric Monitoring Instrument (TROPOMI). Since its launch on 13 October 2017, the S5P mission has been acquiring almost 7 years of nadir ozone profile data, retrieved from the UV spectral bands 1-2 (270–330 nm). The retrieval algorithm of the ozone profile can be strongly affected by systematic effects in the measured radiance, and

absolute calibration of the input spectra is necessary. In this study, we characterize the TROPOMI bands 1-2 radiometric bias in comparison with simulations obtained with the Determining Instrument Specifications and Analysing Methods for Atmospheric Retrieval (DISAMAR) radiative transfer model. To account for these systematic effects, a radiometric correction (soft-calibration) is applied to the input measurements, which results in the reduction of the spectral reflectance fit residuals of 20-30%, improved precision of the integrated total and tropospheric ozone columns of 10-15%and a reduction of along-

track orbit artifacts. Together with the analysis of the in-flight calibration measurements,the soft-calibration correction spectra provide also useful insights for the improvement of the instrument radiometric calibration. Therefore, bands 1-2 measurements have been reprocessed specifically for this study, with improvements regarding the detector straylight and background signal correction algorithms, to investigate the impact of the updates into the TROPOMI bands 1-2 radiometric bias. The new soft-calibration correction spectra, obtained with the reprocessed bands 1-2 measurements, show significantly reduced magnitude

(around 15-20%, especially in band 1) and also less across-track position and spectral/temporal biases. The new soft-calibration will be part of ESA's next official ozone profile algorithm version 2.9.0, coordinated with L1b update to processor version 3.0.0, and used for the second TROPOMI mission reprocessing.

## 1 Introduction

Daily global ozone profile measurements in the stratosphere and troposphere are valuable to atmospheric research. Ozone is one

of the most important components of our atmosphere, as it is a critical stratospheric absorber of the ultraviolet (UV) radiation, and also a strong oxidant in the troposphere, controlling the abundance and distribution of many atmospheric constituents. Ozone received much public attention in the mid-1980s, when its enormous reduction was observed during Antarctic spring, which was due to human-made chlorofluorocarbon (CFC) compounds (Farman et al., 1985). Since then, the recovery of the



so-called ozone hole has been continuously monitored (World Meteorological Organization (WMO), 2022). Moreover, ozone

is also an important air pollutant and the third most important anthropogenic greenhouse gas in the middle-upper troposphere (Bourgeois et al., 2021). To separate dynamic and chemical effects on ozone which vary with altitude, it is not sufficient to limit the measurements to total ozone column amount, but the key of ozone monitoring is the precise measurement of its vertical distribution at high spatial and temporal resolution (Chance et al., 1997). High-quality tropospheric and stratospheric ozone profile measurements are available from ozonesondes and lidar techniques, however their geographical and temporal coverage

is quite limited. For this reason, complementary satellite data provide sufficient spatial and temporal coverage to determine global vertical ozone distributions.

The first theoretical idea of deriving ozone vertical distributions using the back-scattered radiation from the atmosphere using nadir-viewing satellite observations was described in (Singer et al., 1957). Exploiting the low penetration depth of the UV radiation in the atmosphere at short wavelengths, the retrieval of ozone vertical profiles became possible with nadir-

viewing satellite instruments. Since 1970, the launch of the Backscattered UltraViolet (BUV) and Solar Backscatter Ultra Violet (SBUV) instruments allowed systematic measurements of total ozone and ozone profile from space. Using around 10 monochromatic measurement channels in the wavelength range 250-340 nm, these instruments provided more than 40 years of total ozone and ozone profile data record (Bhartia et al., 2013). Since absolute instrument calibration and long-term instrument characterization are needed to produce high quality ozone data, DeLand et al. (2012) developed specific calibration

techniques for the SBUV version 8.6 ozone data product. This method was derived and applied directly at radiance level rather than being determined from ozone values or other retrieved quantities, and it was referred to as "soft" calibration procedure (as distinguished from the "hard" calibration based on laboratory or dedicated in-flight measurements). Moreover, the soft-calibration allowed to evaluate and to support the spectrally dependent calibration adjustments at a level of accuracy greater than what laboratory standards could provide.

In 1995, the launch of the Global Ozone Monitoring Experiment (GOME) instrument allowed the first space-born measurements of contiguous spectra in the UV-visible(VIS) spectral regions. Observing the entire wavelength range between 240-790 nm with a spectral resolution of 0.2-0.3 nm, the GOME instrument made it possible to retrieve the vertical ozone distribution in the stratosphere and in the troposphere, exploiting the 4 orders of magnitude decrease in the ozone absorption cross section in 270–330 nm (Chance et al., 1997). The ability to resolve temperature-dependent spectral structure in the Huggins

band (310–340 nm) allowed the GOME instrument to obtain also the knowledge of the profile information below the ozone peak, an important advance over the SBUV instruments. However, to extract useful tropospheric ozone information from the temperature-dependent Huggins band it was crucial to perform an accurate spectral fit. Together with first GOME ozone profile retrieval using the Optimal Estimation (OE) method (Munro et al., 1998), several studies were published on techniques to improve the absolute radiometric and spectral calibration of the GOME measured spectra, among which also soft-calibration

techniques (e.g. Van Der A et al. (2002); Liu et al. (2005)).

With the subsequent instruments, the SCanning Imaging Absorption SpectroMeter for Atmospheric CHartographY (SCIA-MACHY) on ENVISAT (2002), the Ozone Monitoring Instrument (OMI) aboard the EOS Aura spacecraft (2004), GOME-2 on board the Metop-A,-B,-C satellites (2006) and the Ozone Mapping and Profiler Suite (OMPS) on the SNPP NOAA satellite



(2011), the spatial resolution of the measurements was significantly improved. TROPOMI (Veefkind et al., 2012) is a follow-up

of the SCIAMACHY and OMI series, with an improved spatial resolution. The ozone profile retrieval of TROPOMI uses the spectral bands 1-2 (270–330 nm). The band 1 of the UV detector presents several radiometric calibration challenges (discussed in Sect. 4) but it also contains most of the ozone profile information in the stratosphere. To reduce the radiometric offsets and improve upon the quality of the ozone profile retrieval product in terms of precision and product artifacts (e.g. along-track stripes), the TROPOMI operational ozone profile retrieval algorithm also uses a "soft" calibration correction on the input spec-

tra. The soft-calibration has been indeed the typical approach followed in several scientific ozone profile retrieval, as in Liu et al. (2010) for OMI, Cai et al. (2012) for GOME-2, Bak et al. (2017) for OMPS-NP, and in Zhao et al. (2021) and Mettig et al. (2021) for a scientific TROPOMI product. Other types of vicarious calibration are described for example in the RAL ozone profile retrieval scheme, for the GOME-2 experiment (Miles et al., 2015), where the spectral correction for the instrument degradation is derived from the ratio between climatological modeled reflectances and observed spectra.

This study describes the soft-calibration correction applied to the TROPOMI measurements in bands 1-2 used as input for the operational TROPOMI ozone profile retrieval algorithm, and the soft-calibration spectra improvements in terms of reduced radiometric biases, as a consequence of adjustments of specific Level 0-1B (L0-1B) data processing algorithm steps, regarding the straylight and background signal corrections. Section 2 starts with the description of the UV module measurement system and the L0-1B data processing algorithm steps, focusing specifically on the straylight and background signal correc-

tion algorithms. In Sect. 3, we give a general outline of the ozone profile retrieval algorithm, focusing on the pre-processing steps necessary to prepare the input measurements to the OE fit. Finally, Sect. 4 and 5 show, respectively, the results of operational soft-calibration and the improvements shown by the new soft-calibration, obtained using the reprocessed bands 1-2 measurements.

## 2 TROPOMI UV measurement system

TROPOMI ozone profile retrievals are performed using the radiance and irradiance from the measurements in bands 1-2, which constitute the two halves of the UV spectrometer and cover roughly the spectral range (267–300 nm) and (300–332 nm) (Kleipool et al., 2018). Bands 1-2 are both part of the UVN measurement system, which will be discussed in this section. The description of the optics, detector and electronics of the UVN module is followed by the schematic of the algorithm steps performed by the L0-1B data processor in the inverse model to obtain the geo-located and calibrated radiance, irradiance and

calibration measurements for each spectral band. In this section, we focus on the algorithm developments introduced in the data processor following the update and re-analysis of the in-flight bands 1-2 calibration measurements, mostly regarding the detector straylight and the background signal corrections. The impact of the new developments on the soft-calibration spectra will be discussed in Sect. 5.



## 2.1 UVN instrument module

TROPOMI (Veefkind et al., 2012) is a space-born nadir-viewing push-broom imaging spectrometer on board of the Sentinel-5 Precursor (S5P) satellite launched on 13 October 2017, on a Sun-synchronous low-Earth orbit. The imaging system enables daily global coverage, with a spatial resolution of $5.5 \times 3.5$ km$^2$ in nadir (Ludewig et al., 2020). TROPOMI uses a single telescope to image the target area onto a rectangular slit, which represents the entrance of the spectrometer system. There are four spectrometers in the system, divided into two modules, measuring the medium-wave UV, long-wave UV/visible (UVIS),

near infrared wave (NIR), and short-wave infrared (SWIR) reflectance of the Earth. Each spectrometer images the slit on its own detector, dispersing the light by means of a grating. The UV, UVIS and NIR spectrometers are jointly referred to as UVN, and their output falls into charge-coupled device (CCD) detectors. The SWIR part of the instrument uses instead a complementary metal-oxide semiconductor (CMOS) detector. Each detector is divided into two halves, yielding to a total of eight spectral bands (Kleipool et al., 2018).

The optical layout of the UV spectrometer consists of three lenses, de-centered and tilted with respect to the optical axis in order to get a good co-registration performance and to remove unwanted specular reflections from the system (Babic et al., 2022). In order to reduce the amount of spectral straylight that could reach the detector, a spatially varying coating is used on the flat side of the last surface before reaching the detector. At each location on the lens, the coating transmits light of the expected wavelength, and it reflects light whose wavelength is 15nm larger. At the end of the UV spectrometer, the light

falls onto the CCD detector, which is a 2-dimensional detector with one dimension corresponding to the spatial (across-track) dimension and the other to the spectral (along-track) dimension one. The read-out of the CCD is performed from two different read-out ports defining the bands 1 and 2. Therefore, the two bands originate from the same detector but vary in read-out settings. The read-out of the image in the detector can indeed transfer more lines together (row binning) in the spatial direction (binning in the spectral direction is not possible), so that if the binning factor (i.e. the number of rows that are binned together)

is larger, the read-out noise is suppressed but also the spatial sampling and the data rate are reduced. Moreover, because of the low radiance levels at wavelengths below 300 nm, the binning factors in band 1 are larger than in band 2, in order to increase the signal-to-noise ratio (SNR). The binning factor also varies in the across-track direction, being larger at the center of the swath and decreasing towards the edges in order to reduce the growth of the across-track pixel size. Therefore, the across-track pixel size in band 1 varies between 28km (at the center) and 60 km (at the edges), while varying between 3.5km and 15km in

band 2. The along-track pixel size is instead defined by the integration time and it is the same in the two bands ($\sim$5.5km).

## 2.2 Instrument radiometric calibration

For the accurate retrieval of atmospheric constituents, it is essential that the instrument is well calibrated with respect to known radiometric sources. Before launch, TROPOMI has been extensively calibrated through a series of on-ground calibration measurements which have been used to compute the so-called calibration key data (CKD) (Kleipool et al., 2018). The CKDs

serve as parameters for the correction algorithms applied by the L0-1B data processor in the so-called "inverse model" to provide the absolute geo-located and calibrated radiance and irradiance from the raw instrument data as received from the





ground system. A high-level overview of the different correction algorithms is shown in Figure 1, adapted from (Kleipool et al., 2018). The light blue blocks in the figure refer to generic corrections addressing instrument-wide effects, e.g co-addition

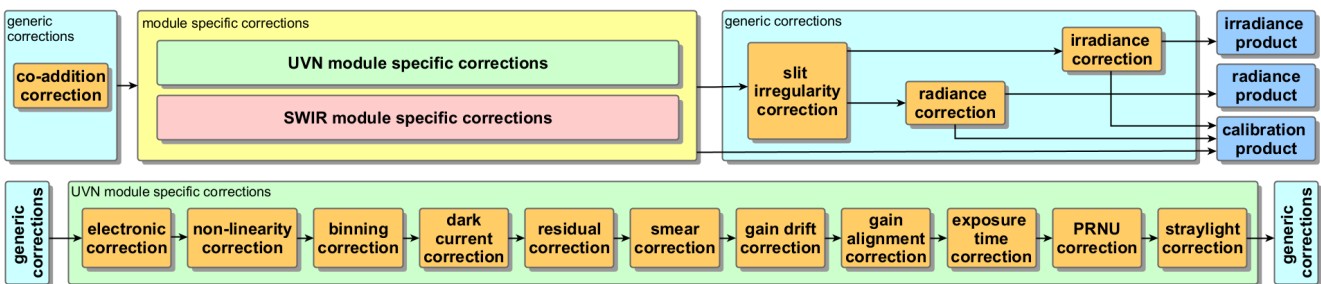

**Figure 1.** High-level overview of the L0-1B processing steps in the whole instrument (upper diagram), and for the UVN module (lower diagram), omitting flagging and annotation corrections. The figure has been adapted from (Kleipool et al., 2018).

or the radiometric irradiance and radiance response, while the yellow block is the module-specific corrections. The lower

diagram shows the UVN-module specific corrections, omitting flagging and annotation corrections. It is worth mentioning that the module-specific corrections include additive corrections, while the generic ones correct exclusively for multiplicative effects.

In-flight measurements are crucial to test and further calibrate the instrument. After launch, TROPOMI was commissioned for 6 months and during this time the instrument calibration has been validated and improved significantly (Ludewig et al.,

2020). In-flight measurements revealed aspects of the instrument, such as temporal drifts in both electronics and optics, that were unforeseen by the inverse model built using the CKDs obtained from the on-ground calibration measurements. In particular, the in-flight radiometric calibration of the UV spectrometer showed inconsistencies with on-ground results and an increase in the radiometric throughput was observed in band 2, possibly due to the reduced efficiency of coatings (the so-called bleaching effects). Figure 2 shows the un-corrected bands 1-2 irradiance measurements for several orbits during the mission,

in comparison to an early orbit (2819, on April 30, 2018). This image shows the strong spectral dependence of the (expected) optical degradation on the UV irradiance measurements, and it reveals the band 2 spectral ageing (310–320 nm), together with unexpected additive effects (associated with incompletely uncorrected for straylight) at the 280 nm and 286 nm peaks, independent from solar variability effects.

Although unforeseen by the model, these effects can be accounted for by introducing additional correction processing steps

or by updating the current CKDs with newly acquired CKDs from the regular in-flight irradiance measurements. Example of these updates in the CKDs are described in the following sections.

### 2.2.1 Straylight correction

The characterization of the detector straylight is also part of to the instrument radiometric calibration. Straylight is generally defined as any light that falls on a detector pixel which by optical design is not intended to detect that light. In an imaging



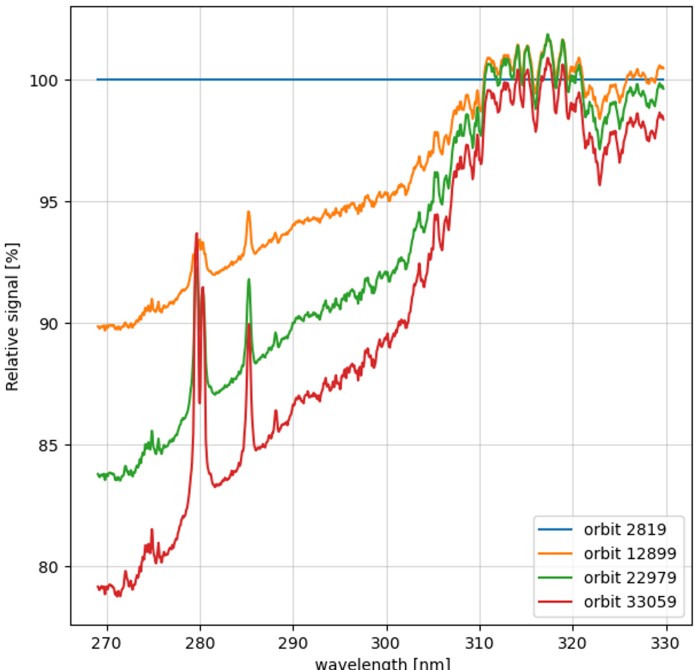

**Figure 2.** The optical degradation of UV irradiance measurements during the TROPOMI mission up to the beginning of 2024 (orbit 33059, on February 2, 2024), in comparison with a reference orbit 2819 on April 30, 2018, (blue horizontal line). The degradation has a strong spectral dependence and it is also partly correlated to the signal magnitude.

spectrometer using two-dimensional detectors like TROPOMI, straylight can become quite complex because light can scatter in both the spectral and spatial dimensions, and the source of the straylight might lie even outside the intended spatial or spectral range. In TROPOMI, straylight is defined in several ways, depending on its origin: out-of-field straylight, which is the light that originate outside the intended spatial field of view of the telescope; out-of-spectral-range or in-band straylight, defining respectively the light which originate outside or inside the intended spectral range of the spectrometer; near-field or far-field

straylight (both in-band), causing spurious signals, respectively, in the proximity or further away (but on the same detector) of the pixels expected to receive the signal; and finally, ghost signals, originating in the optics (Babic et al., 2022). In TROPOMI, all the above listed types of straylight have a spatial and a spectral component when they arrive at the detector, therefore the straylight correction algorithm implements a 2-dimensional correction, which only addresses the in-band straylight. The out-of-spectral-range straylight was found to be significant only for the NIR spectrometer (and, consequently, corrected for)

(Ludewig et al., 2020).



The UVN in-band straylight correction algorithm is implemented using a convolution kernel, which first computes the straylight signal for a given input signal and then it subtracts the calculated straylight signal from the input one. This correction algorithm is implemented as a last step (additive) in the module-specific corrections (Figure 1). The convolution kernel basically represents the straylight response function and it is a CKD parameter derived from the on-ground calibration measurements. The 2-dimensional shape of the straylight convolution kernel is shown in the dashed blue line in Figure 3: (a) in the spatial dimension, (b) in the spectral one.

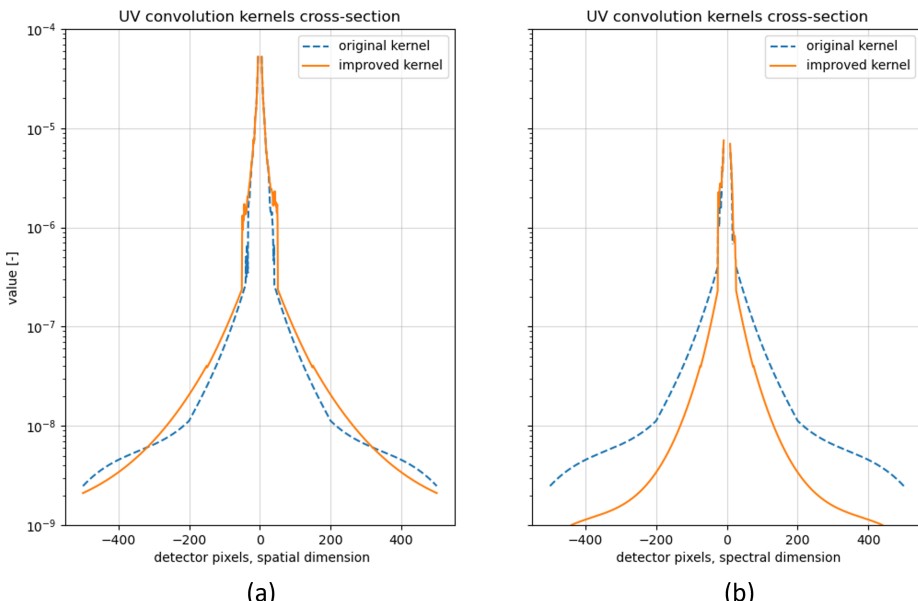

**Figure 3.** The 2-dimensional cross section of the original (blued, dashed line) and the improved (orange, solid line) straylight convolution kernel: **(a)** the detector spatial dimension, **(b)** the detector spectral dimension.

In addition, in-flight measurements also monitor part of the detector straylight. This can be done using the straylight detector rows which are right above and below the directly illuminated region (or science region) in each CCD detector (Figure 4a), so that they are not directly illuminated by the nominal imaging path. Since these rows contain only straylight, it is possible to use these measurements to compare the calculated straylight (from the straylight convolution kernel) with the observed one in the upper and lower straylight region (USLR, LSLR). The comparison is shown in Figure 4b-c. Figure 4b shows the straylight growth in the irradiance measurements of bands 1-2 during the TROPOMI mission until January 2025, obtained from the straylight measurements in the LSLR/USLR (observed, obs) and computed with the straylight correction algorithm (calc LSLR/USLR). Figure 4c shows the difference between observed and calculated straylight for two specific wavelengths: in band 1, at 289 nm, and in band 2, at 321 nm. It is clear from the figures that the straylight correction algorithm underestimates the straylight signal since the beginning of the mission, and that the difference with the measured straylight has increased over



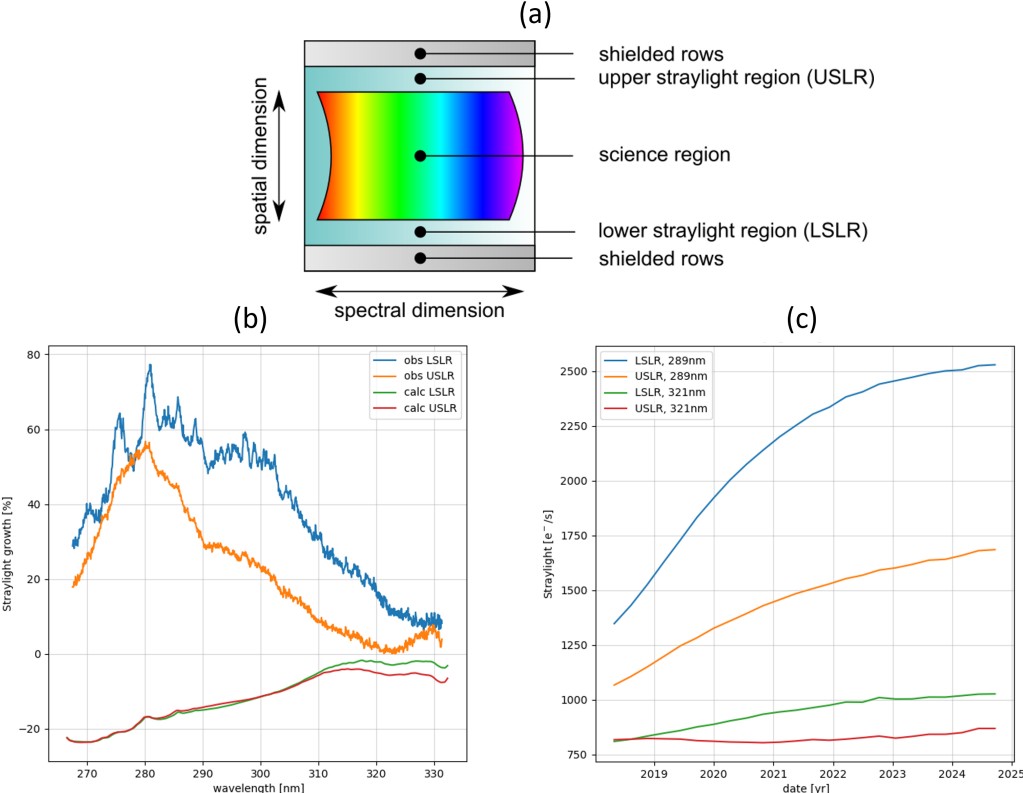

**Figure 4. (a)**: Schematic view of a TROPOMI UV detector. **(b)**: Measured straylight increase in the irradiance since the beginning of the mission (April, 2018) until January 2025, observed and computed (with the straylight correction algorithm) in the LSLR and USLR regions. **(c)** Difference between observed and calculated straylight at two specific wavelengths: at 289, for band 1, and at 321 nm, for band 2. The differences are shown for both the USLR and LSLR detector regions.

time. This large difference suggests two possible solutions: the re-assessment of the straylight convolution kernel shape from the on-ground calibration measurements, and the simultaneous implementation of a "dynamic" straylight correction which removes, for each measurement, a second straylight contribution derived from this difference. These two adjustments result in

a higher total straylight signal which is subtracted from the input signal. It is also important to mention that the implementation of the dynamic correction avoids that the straylight additive increase is accidentally corrected by the degradation correction (causing the "multiplicative" peaks at 280 nm and 286 nm in Figure 2), which would lead to errors in the radiance measurements correction.

The improved version of the straylight convolution kernel shape is shown in the orange solid line in Figure 3. The main

change is a shift in relative weight from the spectral towards the spatial direction to avoid a too large contribution from the far-field part of the kernel, from band 2 towards band 1. Especially for radiance measurements, where the signal magnitude



difference between the bands is even more pronounced, the original kernel would result in a too strong straylight correction. As already mentioned, the original kernel is based on the on-ground straylight characterization of the instrument which was performed with a limited set of lasers (for the near-field straylight) and white light sources (for the far-field component of straylight). Because of limited band-width measurements, the real straylight operator was approximated to a single convolution kernel for both band 1 (<300 nm) and band 2 (>300 nm), a compromise due to the availability of the measurements, given the large radiance signal magnitude differences between bands 1-2. Therefore, the original kernel model shape was updated to take into account this difference, and by implementing an additional dynamic straylight correction to address the discrepancy between observed and calculated straylight, shown in Figure 4c.

## 2.3 Background signal correction

Apart from the actual illumination signal, the signal recorded by the CCD detector consists also of a number of additional components, not all of which can be modeled (Kleipool et al., 2018). The background signal (or residual) correction is intended to correct for the remainder of all these additive signal components which are usually unmodeled artifacts. The correction is a CKD image which varies smoothly in the across-track direction, apart from some discontinuous signal (spikes) and binning scheme artifacts. In most cases, the residual signal (and its uncertainty) is far lower than the measurement itself. Moreover, because of the effort required in computing such a non-constant residual CKD, which depends on several instrument configurations, it was initially decided to not implement it in the data processing (Babic et al., 2022). However, to address remaining observed irregularities (e.g. electronic binning jumps, on one hand, and irregularly behaving pixels due to random telegraph signal (RTS) effects, on the other) which have a significant impact in band 1, because of the very low radiance level at short-UV wavelengths, a sampled background (residual) signal correction was introduced in the L0-1B data processing. It is also important to mention here that the simultaneous observation of spatial artifacts in the operational ozone profile soft-calibration (shown and discussed in Figure 9) helped to focus on the hypotheses of offsets and residual irregularities that were not completely accounted for: a successful example of L1B-Level-2 (L2) interaction.

## 2.4 Discussion

The analysis of in-flight calibration measurements is essential to monitor, improve and adjust the CKDs established during the on-ground calibration campaign of the instrument. Figure 2 shows for example that, in the (expected) optical degradation of the instrument, there exist some high-frequency spectral components at the band 1 Fraunhofer lines, which cannot be explained by the bleaching hypothesis. Part of these irregularities could be attributed to the instability of the stimulus used for the degradation measurements (i.e. the Sun and its long- and short-term solar cycles). However, they strongly suggested the presence at the same time of an uncorrected, monotonously increasing, additive term. Other correction algorithms were also investigated, but as the in-flight and on-ground calibration measurements showed consistence, it was not required to adjust any other CKDs. After discussing the re-analysis of the on-ground and in-flight calibration measurements, yielding the aforementioned correction algorithms, we will now show the effect on the observed bias in bands 1-2 (Veefkind et al., 2021), which is corrected for by applying a soft-calibration correction prior to the ozone profile retrieval algorithm itself.





## 3  Ozone profile retrieval algorithm

The operational TROPOMI ozone profile retrieval algorithm uses the optimal estimation (OE) method (Rodgers, 2000) to derive the ozone concentration as a number density at 33 pressure levels in the atmosphere, using observations in the TROPOMI spectral region from 270–330 nm. The information content of the retrieval is characterized by about five to six vertical sub-columns of independent information, a vertical sensitivity nearly equal to unity at altitudes from about 20 to 50 km, and an effective vertical resolution ranging within 10–15 km, with a minimum close to 7 km in the middle stratosphere (Keppens et al., 2024). Diagnostic information about the ozone profiles and the spectral fit are also provided, such as the averaging kernel matrix, the diagonal elements of the a-priori ozone profile error covariance matrix, and the a-posteriori error covariance matrix (Veefkind et al., 2021). Because the retrieval of height-resolved ozone information from the UV spectrum is a nonlinear under-constrained problem, the OE method solution results from a combination of the measured spectra information with an additional *a priori* ozone profile constraint and its associated errors, to linearize and iterate until convergence is obtained.

Before performing the retrieval itself, it is important to perform a series of pre-processing steps in a well-defined order on the input measurements to obtain the re-calibrated and corrected radiance and irradiance measurements. This section provides an overview of these pre-processing steps, for a comprehensive description of the retrieval algorithm we refer the reader to the Algorithm Theoretical Basis Document (Veefkind et al., 2021).

### 3.1  Measurements pre-processing

A diagram of the pre-processing steps is shown in Figure 5. The first step is the spectral calibration of the irradiance measurements. This is performed by fitting a wavelength shift parameter on the irradiance spectrum in the spectral fit window (270—320 nm) using the precise knowledge of the Fraunhofer lines of a solar reference spectrum. For the Earth radiance spectrum, the assigned wavelengths provided by the L0-1B processor are used.

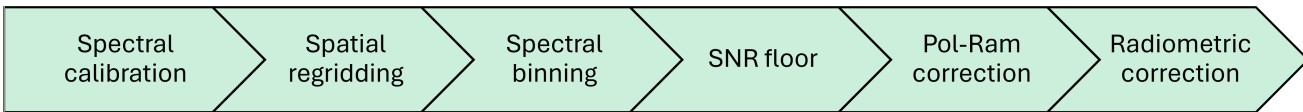

**Figure 5.** The pre-processing steps performed on the measured spectra before the Ozone Profile retrieval algorithm.

As explained in Sect. 2, bands 1-2 are derived from the same UV detector, but have a different detector row binning. To spatially co-register these bands, the band 2 pixels are binned using the same binning scheme as used in the read-out of band 1. As the spectral response varies over the detector, the instrument spectral response function (ISRF) for the binned pixels is also generated. After the co-registration in the across-track direction, five scanlines in the flight direction are also averaged, resulting in a continuous radiance spectrum in the fit window (270–330 nm), for 77 across-track ground pixels, with spatial resolution of $28 \times 28$ km$^2$ (across-track x along-track) in nadir after August 6, 2019 and $28 \times 35$ km$^2$ before this date.





To reduce the number of line-by-line radiative transfer computations on the spectral grid of the measured input spectra, three pixels are averaged in the spectral direction (spectral binning) to reach an oversampling ratio of at least 2.3 (Veefkind et al., 2021). The signal to noise (SNR) and the ISRF also updated to take into account this spectral binning. Moreover, the final SNR is clipped to a maximum of 150 (noise floor) in order to improve the convergence of the algorithm.

To compute the modeled reflectances for the OE method, the DISAMAR radiative transfer model (RTM) (de Haan et al., 2022) is used. The DISAMAR RTM can be run in vector mode including polarization, however in order to reduce the retrieval computational time, the RTM is run in scalar mode for elastic scattering in the operational ozone profile algorithm. Therefore, a correction for Raman scattering and polarization has been developed and applied to the input measured spectra before the retrieval itself. The correction is based on a large dataset which is used to train a neural network to obtain the correction

factor parameters from the comparison of spectra computed with and without polarization and Raman scattering. Finally, the correction factor parameters are given and applied as a function of the wavelength, sun-satellite geometry, surface albedo, pressure and total ozone column (Veefkind et al., 2021).

The last step of the pre-processing is the radiometric (or soft-calibration) correction, described in the following sections.

## 4 TROPOMI UV soft-calibration

The main keys to successful ozone profile retrievals are accurate calibration and simulation of the measurements. TROPOMI bands 1-2 have been extensively analysed and calibrated before and after launch (Kleipool et al. (2018); Ludewig et al. (2020)). However, large systematic radiance differences are still present when compared to forward model calculations, especially in band 1, and they are addressed by the soft-calibration correction. In this section we describe the soft-calibration correction, an essential correction step complementary to the instrument radiometric calibration, used spefically for TROPOMI spectral

bands 1-2. This soft-calibration version is in operation in the ozone profile algorithm version 2.8.0., activated in November 2024. For details on the different processor versions, the reader is referred to the product read-me file (PRF), available online (http://doi.org/10.5270/S5P-j7l9xvd).

Figure 6 shows the effect of the soft-calibration on the ozone profile retrieval (orbit 19452, on July 15, 2021). The residuals of the reflectance fit for several across-track positions, respectively, without and with soft-calibration are shown in Figure

6(a-b). Residuals are overall larger in band 1 than in band 2, considering however that the absolute intensities in band 1 are smaller by order of magnitudes with respect to band 2. The largest peaks in the residual are in the stronger solar absorption lines (Mg, Fraunhofer lines), where the radiance is very small. After the correction, the residuals decrease and the across-track position uniformity improve. The retrieved ozone total column and 0-6km integrated sub-column are shown in Figure 6(c-d) and (e-f), while their $1\sigma$ standard deviation (indicated as "precision", for consistency with the nomenclature of the TROPOMI

ozone profile product), is shown in Appendix B1. When applying the soft-calibration correction, the maps show improvement in terms of reduction of retrieval artifacts (along-track stripes) and precision reductions of around ~10-15%.

The TROPOMI operational soft-calibration is computed following a similar approach as described in Mettig et al. (2021), characterizing the differences between measured and modeled radiances to compute the correction parameters. To implement a





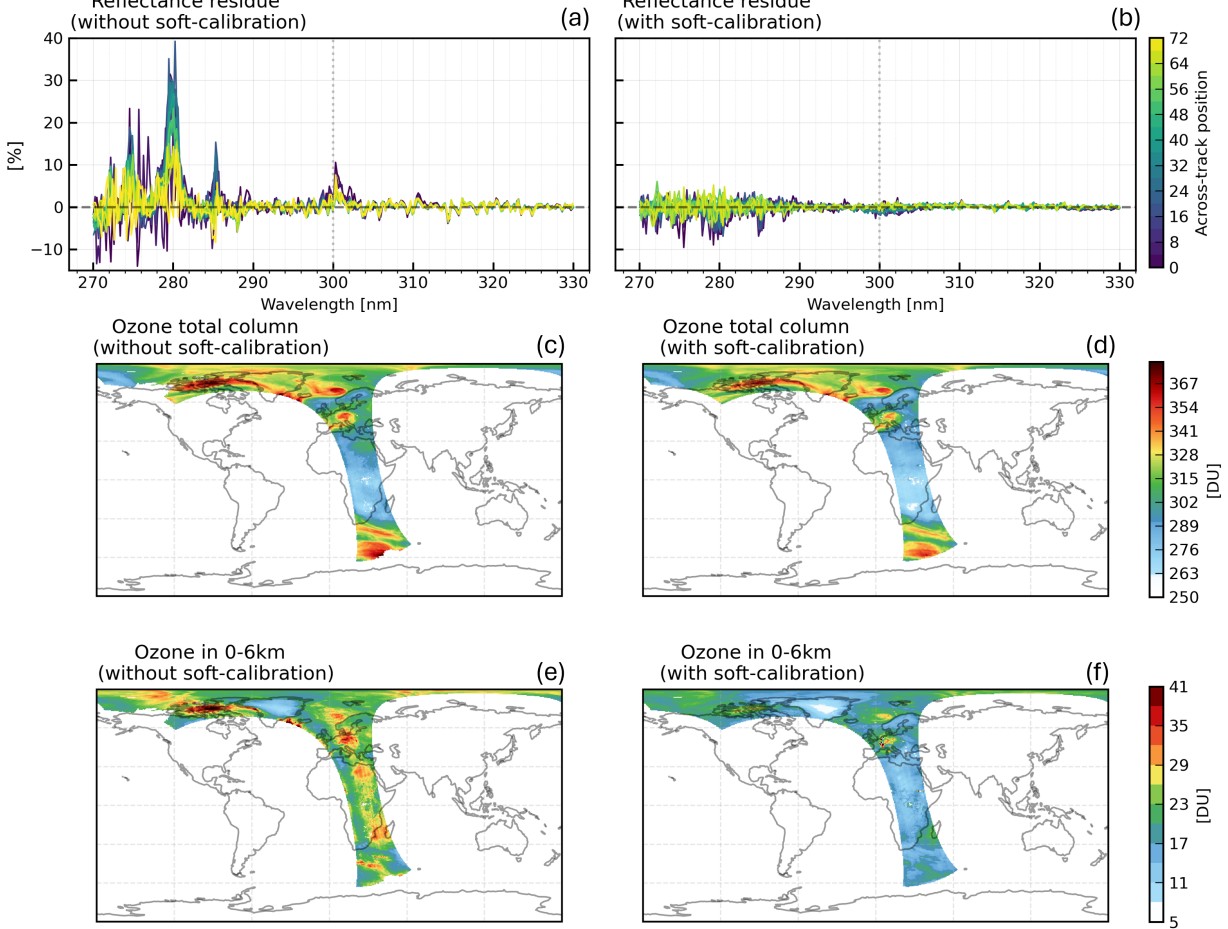

**Figure 6. (a)-(b)**. Reflectance fit residue $\left(\frac{\text{TROPOMI}-\text{model}}{\text{model}}\right)$ of bands 1-2 measurements in the tropics (4°N-9°N), for the TROPOMI Ozone Profile retrieval of orbit number 19452 (15, July 2021), for several across-track positions, without (left) and with soft-calibration (right). **(c)-(d)-(e)-(f)**. Retrieved ozone total and 0-6km integrated sub-column, without (left) and with (right) applying the soft-calibration for the same orbit. The precision of the respective integrated columns is shown in Appendix B1.

consistent correction over the whole mission (2018-2024), a total of 26 TROPOMI L1b orbits have been chosen from bands 1-2

measurements, always over the Pacific Ocean. Four orbits per each year have been chosen to capture an average yearly trend, while results from the comparison are collected per each year altogether and applied on the input measurement accordingly, as described in this section. The chosen orbits numbers are listed in Table A1 in appendix A, with the columns showing the radiance and irradiance orbits, the radiance observation date, and the combined orbit number and date use for the operational implementation of the correction. In this section, the method and the results of the operational soft-calibration are described.





### 4.1 Forward model

The modeled radiances are computed using the DISAMAR RTM using 8 streams (number of Gaussian division points used for integration over polar angles), in accordance with the number of streams used in the ozone profile retrieval algorithm (Veefkind et al., 2021). Since the soft-calibration is performed after the polarization-Raman correction step (Figure 5), the RTM for the modeled radiances is also run in scalar mode, and the measured spectra are corrected for these two effects in the same way as in the retrieval algorithm before the comparison. Pressure, temperature and ozone profiles are obtained from the Copernicus Atmospheric Monitoring Service (CAMS), using the operational atmospheric model forecast. A 3-hours forecast is used, read from the midnight of the observation date (8 timestamps). The global grid has a $0.4 \times 0.4$ degrees, while the vertical grid uses 60 levels before July 9th, 2019, while 137 levels afterwards. The CAMS ozone profiles are scaled to match the TROPOMI Total Ozone (Spurr et al., 2022) and the MLS ozone profile version 5.0 (Livesey et al., 2020). It is important to specify that the modeled atmosphere does not contain clouds or aerosols, but their effect is compensated by adjusting the surface albedo. Therefore, the scene albedo is fitted in a small spectral window (328–330 nm) and assumed to be representative for the entire fitting window. The scene albedo is applied before the polarization-Raman correction, and it is therefore performed with the RTM in vector mode, thus accounting for linear polarization.

### 4.2 Method

After obtaining the modeled spectra for all the across-track positions of the selected orbits, the absolute residuals (measurement - model) are calculated. To compute the correction parameters, the absolute residuals are binned into 20 equally populated percentiles bins (each bin covers 5% of the sorted dataset) based on the radiance, per each across-track position and wavelength. In particular, having the correction as a function of the radiance level enables accounting for both additive and multiplicative errors in the instrument forward model (see Sect. 2.2). After the binning, a wavelength grid of 1200 points is used to group the residuals, covering the spectral fit range (270–330 nm). The mean radiance, the median and standard deviation residuals are computed at the mid-point of each radiance bin, and a 3rd degree polynomial function is fitted through these mid-percentiles points, to ensure a smooth behavior of the correction as a function of the radiance. A representation of the correction is shown in Figure B2 in Appendix B, for the same across-track position 35 and three wavelengths 285 nm, 300 nm and 310 nm, in 2021. An overview of the yearly residuals is shown in Figure 7 for the same across-track position number 35 and two wavelengths (band 1 in Figure 7a, band 2 in Figure 7b). Each line represents the combined absolute spectral radiance residual per each year of the TROPOMI mission. In band 1, the absolute residuals increase over time, while in band 2, the difference among the years is almost negligible. Units are expressed in more traditional units, instead of the SI unit of moles of photons.

### 4.3 Results

The soft-calibration correction is applied operationally by subtracting the correction bias from the uncorrected radiance signal ($R_{\mathrm{corr}} = R_{\mathrm{uncorr}} - \mathrm{correction}$). Therefore, an overly negative correction value means that the uncorrected radiance signal has been largely underestimated. The overview of the operational soft-calibration correction spectra ($\frac{\mathrm{correction}}{R_{\mathrm{uncorr}}}$) is shown in Figure





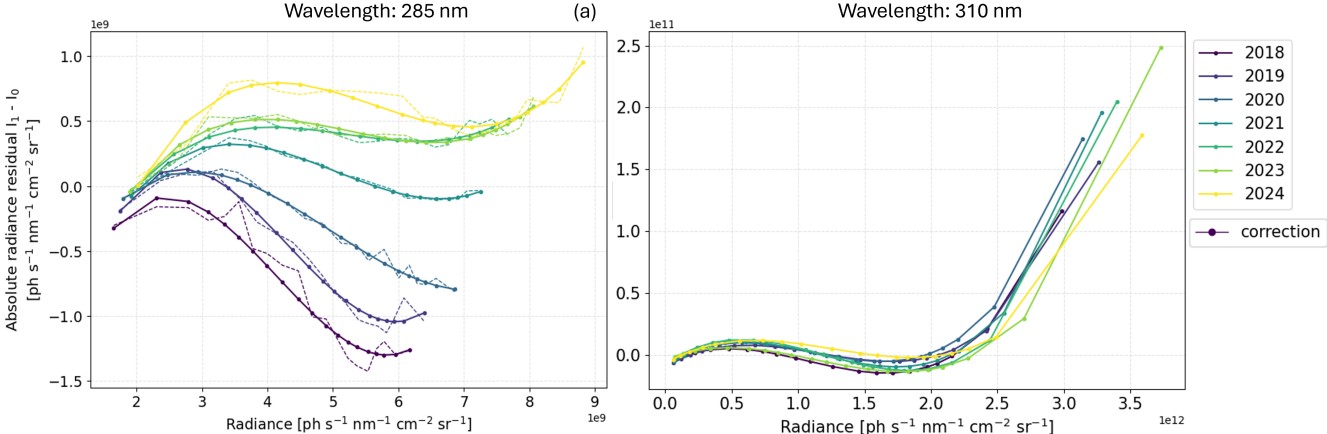

**Figure 7.** Absolute spectral radiance residuals of the TROPOMI operational soft-calibration for the across track position number 35 and two wavelengths: at 285 nm in **(a)**, and at 310 nm in **(b)**. Each line refers to one year of the TROPOMI mission.

8. Figure 8 (a-c), (d-f) and (g-i) show the correction, respectively, at a high, middle and low radiance level, for three different across-track positions (0, 30, 76). The different radiance levels are an indication of the brightness of the atmospheric scenes under investigation, with the low radiance level representing dark scenes, and high radiance bright scenes. An indication of the

radiance levels for orbit 22992 on March 22, 2022 and three ground pixels (0, 30, 76) is shown in Appendix B3, for both bands 1-2.

The correction spectra show a unique spectral shape, with large absolute values in band 1, especially high peaks in the stronger solar absorption lines and at the interface between the two bands (∼300 nm). There are two main characteristics to notice: a concave negative shape between 280–300 nm and the sign change in the stronger band 1 solar absorption lines, mostly

after 2020. It is important to mention that it is especially difficult to find the point in time in which this happens, as the soft-calibration calculations are not computed per each single TROPOMI orbit because it would be very computationally expensive. The correction shows large values (> 40%) at the lowest radiance level (Figures 8 (g-i)), especially for the across-track positions at the edge. The signal in those regions is extremely low and especially difficult to model. Little or zero correction can be seen in band 2, especially for a middle radiance level (Figures 8 (d-f)), while at the highest radiance level (Figures 8 (a-c)), the

correction in band 2 shows larger deviations, especially at the beginning of the mission (blue line, 2018) and at the East-side of the swath. As we already noticed in Figure 7, the difference among the years is larger in band 1 than in band 2. Focusing on band 2, a distinction can be seen in the temporal trend, depending on the radiance level. The highest radiance level (Figures 8 (a-c)) shows an "inverted" temporal trend, with the size of the correction decreasing in time instead of increasing (as in e.g., Figure 8e).

The radiance residuals computed using the soft calibration apporach are expected to come from the TROPOMI measurements (Veefkind et al., 2021), but it can be difficult to distinguish instrumental from model errors. Since the discussion regard-



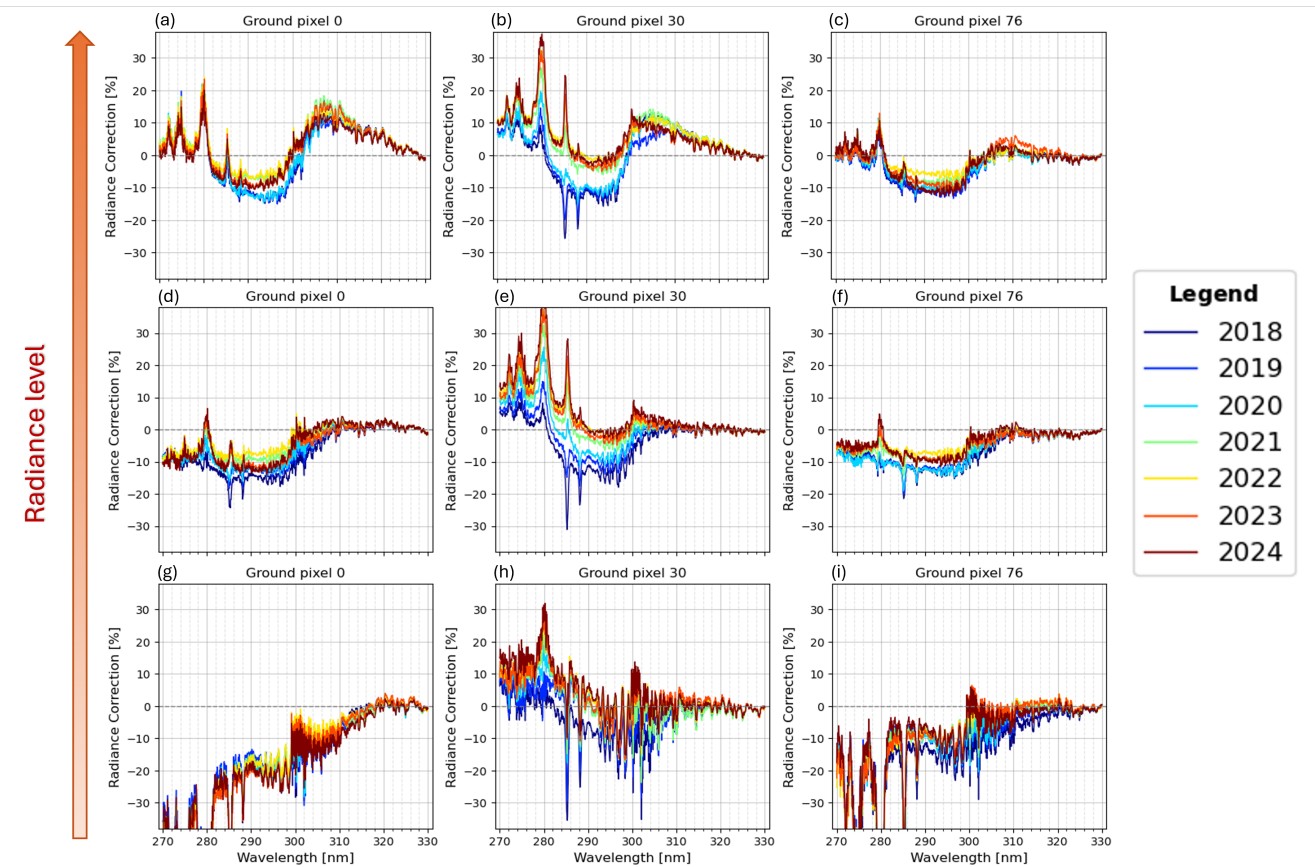

**Figure 8.** The soft-calibration correction spectra $\left(\frac{\text{correction}}{\text{R}_{\text{uncorr}}}\right)$ per each year of the TROPOMI mission. Figure 8 **(a-c)**, **(d-f)** and **(g-i)** show the correction, respectively, at a high, middle and low radiance level, for three different across-track positions (0, 30, 76).

ing the model errors is outside the scope of this study, in the following Sect. 5 we will focus on the impact of the instrumental updates introduced in the L0-1B data processor (described in Sect. 2.2), on the operational soft-calibration correction spectra.

## 5   Improvements

The soft-calibration spectra shown in Figure 8 show a strong spectral, radiance and temporal bias. In this section, we show the impact of the L0-1B data processing updates described in Sect. 2 on the soft-calibration correction and on a single retrieved ozone profile orbit. TROPOMI bands 1-2 radiances have been reprocessed with three adjustments - the improved straylight convolution kernel shown in Figure 3, the dynamic straylight and the background signal correction algorithms -, and the absolute residuals of the soft-calibration are computed using the bands 1-2 reprocessed with a single change in the L0-1B

data processing. The soft-calibration procedure is instead kept the same as described in Sect. 4.2. In this section, the improved spectra and the impact on the ozone profile retrieval will be shown.





Figure 9 shows the result of the different tests, with each panel focusing on a specific soft-calibration case, for all the across-track indices and two orbits: orbit 4889 (on September 22, 2018) on the left, orbit 22992 (on March 22, 2022) on the right. Figures 9a-b show the operational soft-calibration (same spectra as in Figure 8), but for all the across-track indices and

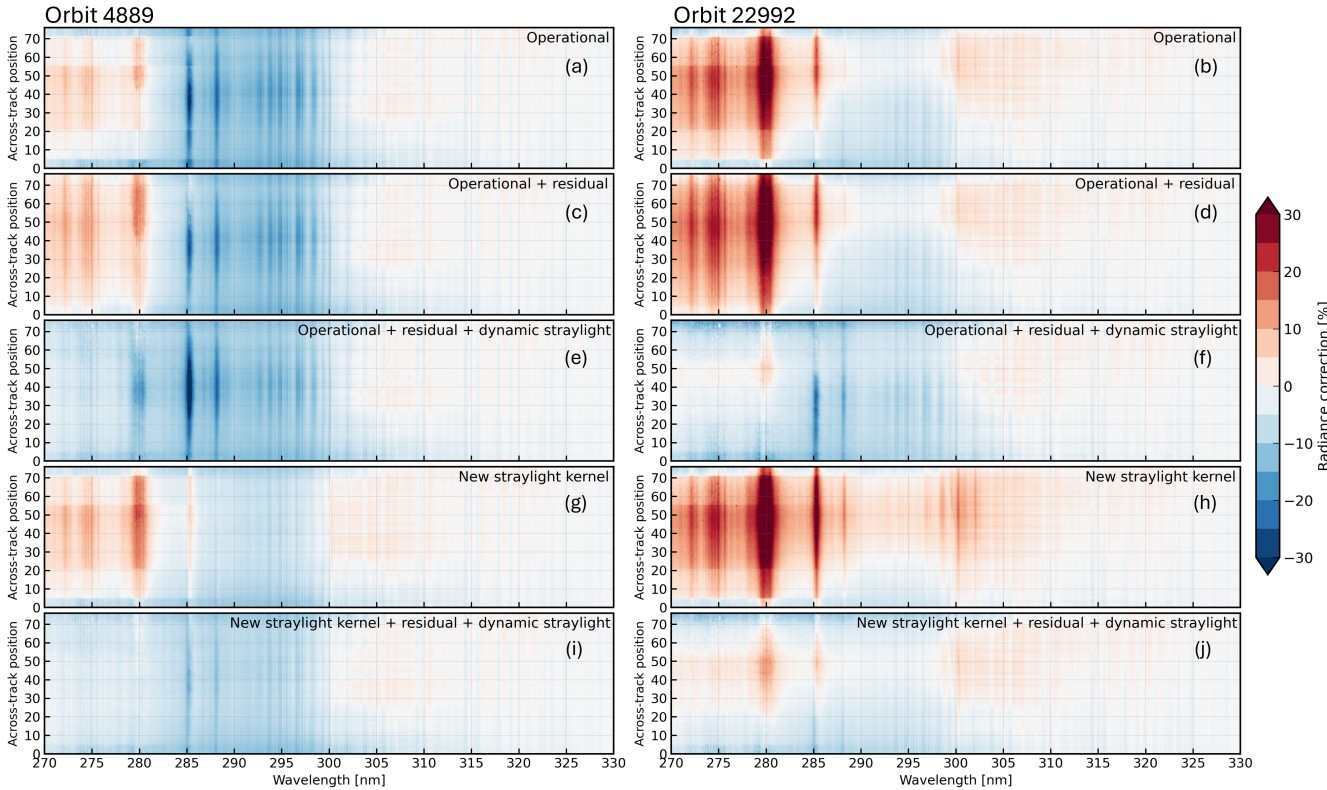

**Figure 9.** Soft-calibration spectra panel for all the across-track positions for orbits 4889 (on September 22, 2018) and 22992 (on March 22, 2022). The different sub-plots show the comparison of the correction spectra produced using different processing of the bands 1-2 radiances: **(a)-(b)** shows the operational case, **(c)-(d)** the operational case with the implementation of the background (or residual) signal correction, **(e)-(f)** implements the dynamic straylight correction algorithm to the case **(c)-(d)**, **(g)-(h)** shows the result with the improved straylight convolution kernel only, **(i)-(j)** implements the background signal correction, the dynamic straylight correction and the improved straylight convolution kernel.

a middle radiance level. These two panels are compared with similar panels in each row, showing a single change in the L0-1B data processing. The first change is the implementation of the background (or residual) signal correction (Figure 9c-d), which has a significant impact on the across-track position artifacts (especially in band 1), resulting in a smoother spatial distribution of the correction. Figure 9e-f show the result of the implementation of the dynamic straylight correction algorithm on the top of Figure 9c-d. These panels show a reduction of the soft-calibration spectra size with respect to Figure 9a-b, 350 but also an overall decrease of the temporal differences between the two orbits. It can be noticed however that the temporal



bias is still present, as the correction is based on the measured straylight in the USLR/LSLR, adjacent to the science region (Figure 4), which might not fully represent the straylight behaviour inside the science region. In Figure 9g-h, the improved

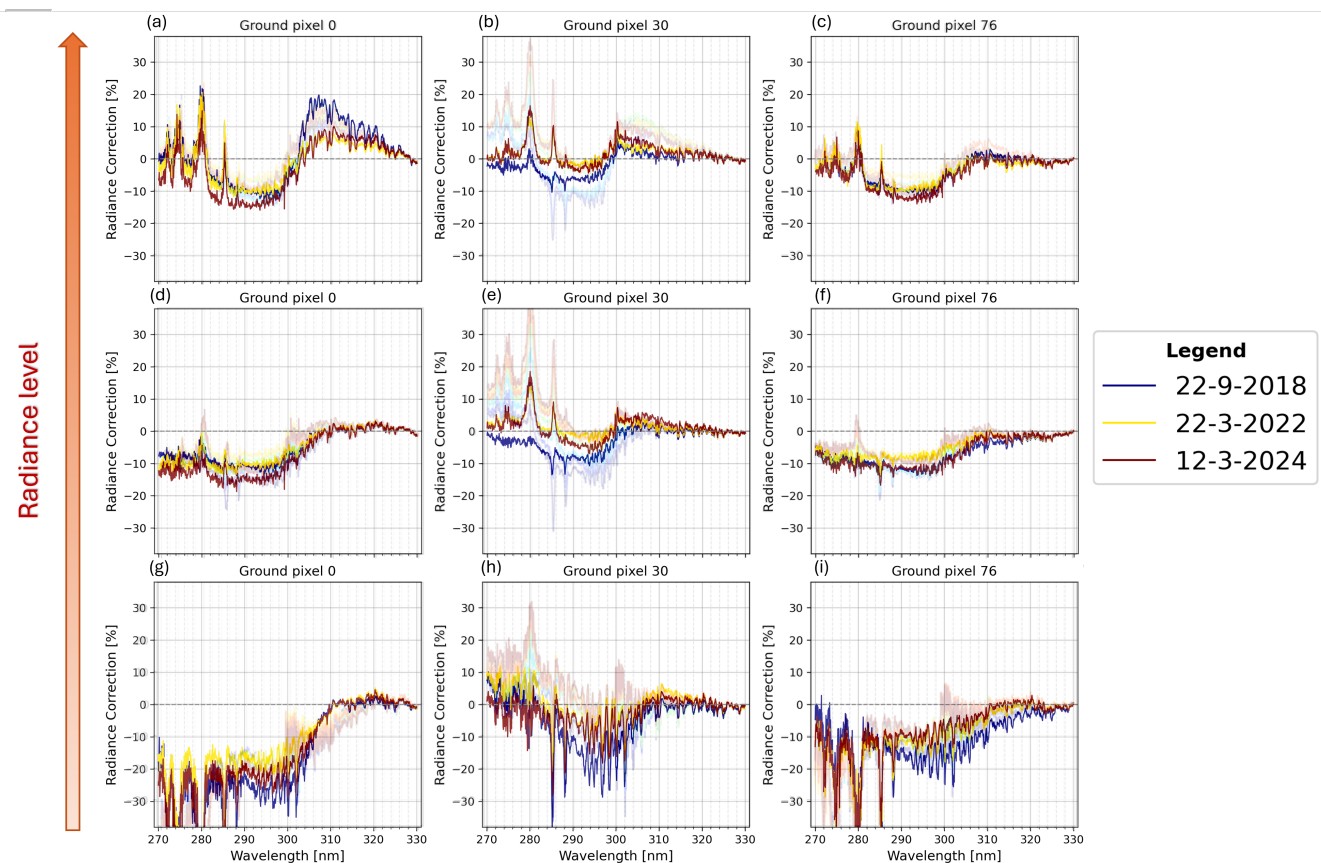

**Figure 10.** The updated soft-calibration correction spectra per each test orbit in 2018, 2022, 2024, for three different across-track positions (0, 30, 76), and a low **(a-c)**, middle **(d-f)**, and high **(g-i)** radiance level. For comparison, the original correction spectra shown in Figure 8 are plotted as semitransparent curves.

straylight convolution kernel is implemented instead of the original one in Figure 9a-b. As expected, the spectral far-field cut off of the straylight convolution kernel mostly affects band 1, especially in the spectral range (285–300 nm). This avoids that

straylight contributions from band 2 would be wrongly assigned to band 1 leading to large correction contributions, especially in the stronger solar absorption lines and early in the mission, leading to larger systematic effects in the optimal estimation retrieval. Moreover, the absolute differences between the orbits is smaller. Finally, Figure 9i-j show the combination of the improved straylight convolution kernel with the dynamic straylight and the background signal corrections. In this panel, the soft-calibration spectra look spectrally and spatially smoother than in Figure 9a-b, but also in comparison with the same

combination however using the operational straylight kernel (Figure 9e-f). The size of the spectra decreases of around 15-20% with respect to the operational soft-calibration, depending on the radiance level and the across-track position.





The temporal variation of the new soft-calibration spectra (Figures 9i-j, obtained with the L0-1B data processing using the three adjustments listed at the beginning of the section) is shown in Figure 10, together with the operational spectra (in transparency, for comparison). The temporal increase from the beginning of the mission is reduced around 30%-40% in the
stronger band 1 solar absorption line, especially for a central across-track position (Figure 10e).

## 5.1 Retrieval results

In this section, we compare the ozone profile retrieval using the operational soft-calibration correction with the retrieval using the new soft-calibration (Figures 9i-j). It is important to notice that, since we are looking at the retrieval of single orbits, we performed the soft-calibration correction on the orbit itself, which is not the procedure followed operationally (as discussed in
Sect. 4.2). In this section, the "new" soft-calibration refers to the spectra shown in Figures 9i-j, while the operational spectra are shown in Figures 9a-b.

Figure 11 presents the zonal mean of several retrieved quantities of the Ozone Profile data product (orbit 22992, on March 22, 2022). Figure 11a-c show the total integrated ozone vertical column using, respectively, the operational (a) and the new soft-calibration correction (b), and their relative difference (c), which are overall small. The black thicker lines in Figure 11c
represent the comparison with the TROPOMI L2 Total Ozone product (Spurr et al., 2022), similar in the two processing, with slighly better comparison using the new soft-calibration for mid-latitudes 20°N-60°N. Looking at the integrated ozone sub-column from 0 to 6km and its precision in Figures 11d-f and g-i, the differences between the two processing are around 3-4%, with higher values using the new soft-calibration. The root-mean-squared (RMS) of the spectral fit and the degrees of freedom (DOF) are shown in Figures 11j-l and m-o. The RMS decreases using the new soft-calibration of around 15-20%,
and the DOF slightly improves in mid-latitudes, both in the south and north hemisphere. The light blue areas show the $\pm\sigma$ standard deviation of the zonal mean. On average, the standard deviation of the retrieved quantities obtained with the new soft-calibration is smaller than with the operational soft-calibration, as can be seen from the mean and standard deviation in Figure C1 (Appendix C).

The ozone profile zonal mean and its precision with both processing is shown in Figure 12, for one across-track index.
Differences are overall smaller than 20%, however larger in the stratosphere, especially at high-latitudes (>60°N), and in the troposphere (0°N-10°N). In Figure 13, we have also compared a single TROPOMI ozone profile with the closest ozone profile retrieved from OMPS-Nadir Profiler (Kramarova, 2017), in five latitude bands. The OMPS-NP profile is shown with the black thick line, while the TROPOMI profile with operational (new) soft-calibration is shown in blue (orange). The dashed lines refer to the secondary upper x-axis, showing the relative difference between OMPS-NP and TROPOMI. The two profiles differ
mostly in the troposphere at north and south mid-latitudes (Figure 13b and d), and in the stratosphere in the south hemisphere (Figure 13a-b). The "new" profile shows equal or less difference with the OMPS profile especially at $\sim 30°$S and $\sim 30°$N.

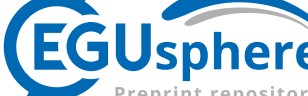



**Figure 11.** Zonal mean of several retrieved ozone profile quantities, using the operational (left column) and the new (central column) soft-calibration correction. The right column shows the relative difference between the two. Several quantities of interest are compared: **(a)-(c)** show the ozone total vertical column, **(d)-(e)** and **(g)-(i)** show respectively the ozone 0-6km sub-column and its precision, **(j)-(l)** show the RMS of the spectral fit, and **(m)-(o)** show the DOF of the ozone. In **(c)**, the black lines show the comparison with the TROPOMI Total ozone product. The light blue areas area the $\pm\sigma$ standard deviation of the zonal average of the variable.



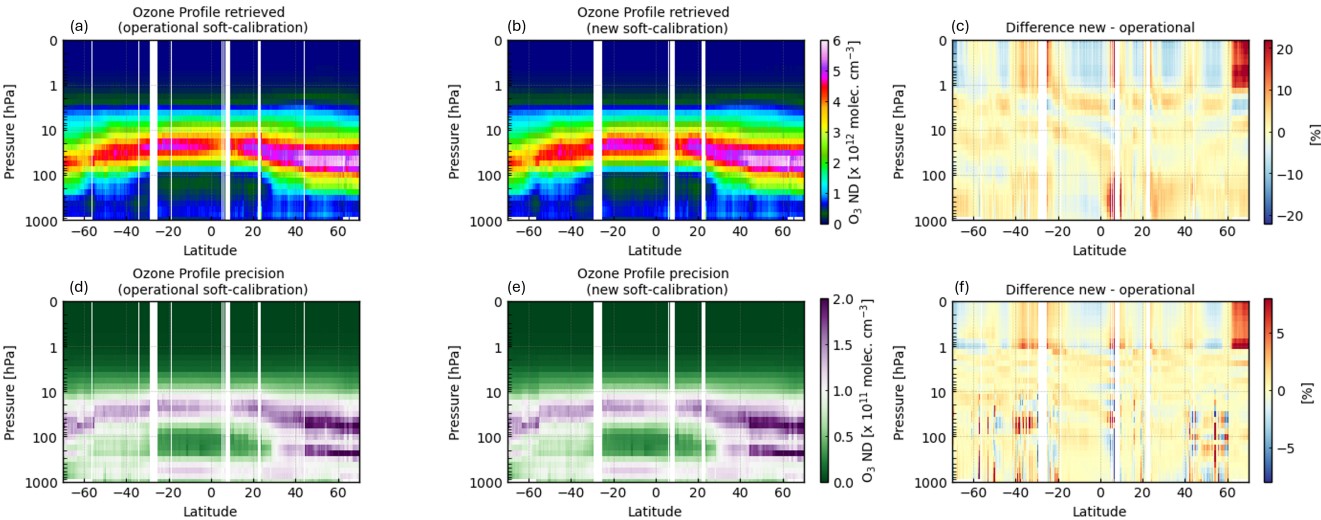

**Figure 12.** Ozone profile zonal mean for across-track position 30 (orbit 22992, on March 22, 2022), processed using the operational **(a)**, and the new **(b)** soft-calibration. The profile precision is shown in **(d)**-**(e)**. The relative difference is shown in **(c)** and **(f)**.

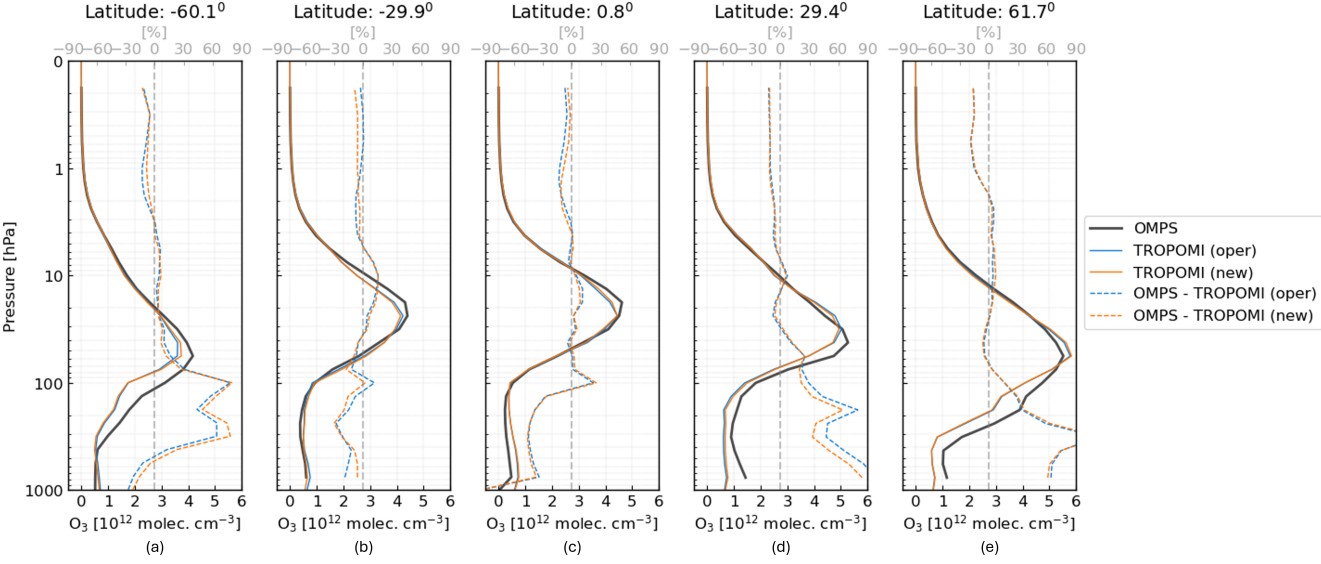

**Figure 13.** Ozone profile comparison between OMPS-NP (black line) and TROPOMI (blue line, operational correction, orange line, new soft-calibration) at several latitudes (**(a)**-**(e)**). The secondary upper x-axis shows the OMPS-NP/TROPOMI difference (dashed lines).



## 6 Conclusions

This work reports on the characterization of the soft-calibration correction used in the TROPOMI operational ozone profile retrieval algorithm, and the improvements obtained in the reduction of its spectral, radiance and temporal biases using the

reprocessed TROPOMI bands 1-2 measurements. The reprocessed bands 1-2 have been updated with adjustments regarding an improved version of the 2-dimensional straylight convolution kernel, the implementation of the dynamic straylight based on the straylight measurements in the upper/lower straylight region (USLR)/(LSLR) of the UV detector, and the implementation of the background signal correction algorithms. The soft-calibration correction is obtained from the computation of the comparisons between modeled and measured radiances, in the spectral range of the retrieval (267–330 nm). Comparisons are averaged over

the seasons to obtain a yearly correction which is applied on the input measurements of the single retrieved ozone profile orbit. The soft-calibration correction reduces the bands 1-2 spectral fit residuals of ∼20-30% and it improves the retrieved integrated ozone total and sub-column in 06km both in terms of retrieval artifacts (along-track stripes) and precision (decrease of ∼15-20%).

The analysis of the soft-calibration correction spectra ($\frac{\text{correction}}{\text{R}_{\text{uncorr}}}$) shows large spectral, across-track position, radiance level

and temporal variation (especially in band 1), which can result in large systematic errors in the estimate of the retrieved quantities, in both stratospheric and tropospheric ozone. The new soft-calibration spectra obtained with the reprocessed bands 1-2 measurements show several improvements with respect to the operational spectra: the large across-track position and spectral biases decrease significantly with the new soft-calibration (around 15-20%) and the temporal variation is also strongly attenuated due to the implementation of the dynamic straylight correction algorithm.

The effect of the new soft-calibration has also been tested on a single orbit of the ozone profile retrieval. The processing with the new soft-calibration shows improvements in terms of reduction of the RMS spectral fit, a slight improvement of the total DOF, and more stability of the retrieved integrated columns (total and 0-6 km sub-column) and its precision. Looking at the zonal mean of the ozone profile, the differences between the processing are around 15-20% and mostly in the stratosphere. Additionally, the comparison of a single TROPOMI ozone profile with the OMPS-NP ozone profile, in five latitude bands,

showed that the new soft-calibration has also an impact in the troposphere, showing reduced differences between the two profiles at ∼ 30°S and ∼ 30°N.

The new soft-calibration will be part of the ESA's next official ozone profile algorithm version 2.9.0, which is expected to be activated in the public data stream in fall 2025, together with the update of the L01B processor to version 3.0.0. The development of the new soft-calibration is therefore a successful example of the interaction between L1B-L2 work. With the

availability of more test data in the next months, it will be possible to extend the current analysis as a preparation of the second TROPOMI mission reprocessing.

*Author contributions.* SDP performed the formal analysis on the soft-calibration spectra initiated and coordinated by EL, EvdP, AL and PV. PV developed the initial soft-calibration algorithm routine. EL is the responsible of the straylight correction algorithm and conceived the improvements together with SDP, EvdP, EvA, MS, AL and PV. EvdP coordinated and developed the different correction algorithms in the



L01B data processing. EvA developed and implemented the dynamic straylight correction algorithm. MvH developed and implemented the background signal correction algorithm. MS and MTL developed and supported the ozone profile retrieval algorithm. SDP and EL conceived and prepared the data visualization. AK is the responsible of the operational Ozone Profile data product validation and contributed to the discussion and review of the manuscript. All authors have revised and commented on the paper.

*Competing interests.* The authors declare that they have no conflict of interest.

*Acknowledgements.* This work has been funded by the Netherlands Space Office (NSO), as part of the TROPOMI Science Contract and by the ESA/Copernicus Atmospheric Mission Performance Cluster (ATM-MPC). The TROPOMI payload, on board of Sentinel-5 Precursor (S-5P), is a joint development by the ESA and the NSO. S-5P is an ESA mission implemented on behalf of the European Commission. The authors express their thanks to the S5P-PAL (Product Algorithm Laboratory) cloud platform of S[&]T's, for providing the technical infrastructure to support the development of the operational soft-calibration correction. This publication contains modified Copernicus S-5P
data, processed by KNMI.



**Appendix A: List of the L1b orbits (bands 1-2) for the operational soft-calibration computation**

| Radiance orbit | Irradiance orbit | Radiance observation date | Combined orbit | Combined orbit date |
|---|---|---|---|---|
| 2818 | 2818 | 30-4-2018 | 4336 | 15-8-2018 |
| 3966 | 3958 | 19-7-2018 | | |
| 4889 | 4888 | 22-9-2018 | | |
| 5713 | 5713 | 20-11-2018 | | |
| 7429 | 7438 | 21-3-2019 | 9344 | 2-8-2019 |
| 9004 | 9013 | 10-7-2019 | | |
| 10082 | 10093 | 24-9-2019 | | |
| 10891 | 10903 | 20-11-2019 | | |
| 12650 | 12658 | 23-3-2020 | 14536 | 2-8-2020 |
| 14197 | 14203 | 10-7-2020 | | |
| 15260 | 15253 | 22-9-2020 | | |
| 16084 | 16093 | 20-11-2020 | | |
| 17815 | 17818 | 22-3-2021 | 19715 | 2-8-2021 |
| 19375 | 19378 | 10-7-2021 | | |
| 20439 | 20443 | 23-9-2021 | | |
| 21262 | 21268 | 20-11-2021 | | |
| 22979 | 22978 | 21-3-2022 | 24396 | 5-8-2022 |
| 24554 | 24553 | 10-7-2022 | | |
| 25660 | 25663 | 26-9-2022 | | |
| 26540 | 26548 | 27-11-2022 | | |
| 28030 | 28033 | 12-3-2023 | 30057 | 1-8-2023 |
| 29875 | 29878 | 20-7-2023 | | |
| 30684 | 30688 | 15-3-2023 | | |
| 31619 | 31618 | 20-11-2023 | | |
| 33335 | 33328 | 20-3-2024 | 34158 | 17-5-2024 |
| 34996 | 35008 | 15-7-2024 | | |

**Table A1.** List of the TROPOMI L1B orbits (bands 1-2) used for the computation of the operational soft-calibration correction parameters.

This soft-calibration is currently in operation in the Ozone Profile processor version 2.8.0, activated in November 2024.





## Appendix B: Soft-calibration

### B1    Effect of soft-calibration on ozone total and 0-6km integrated sub-column precision

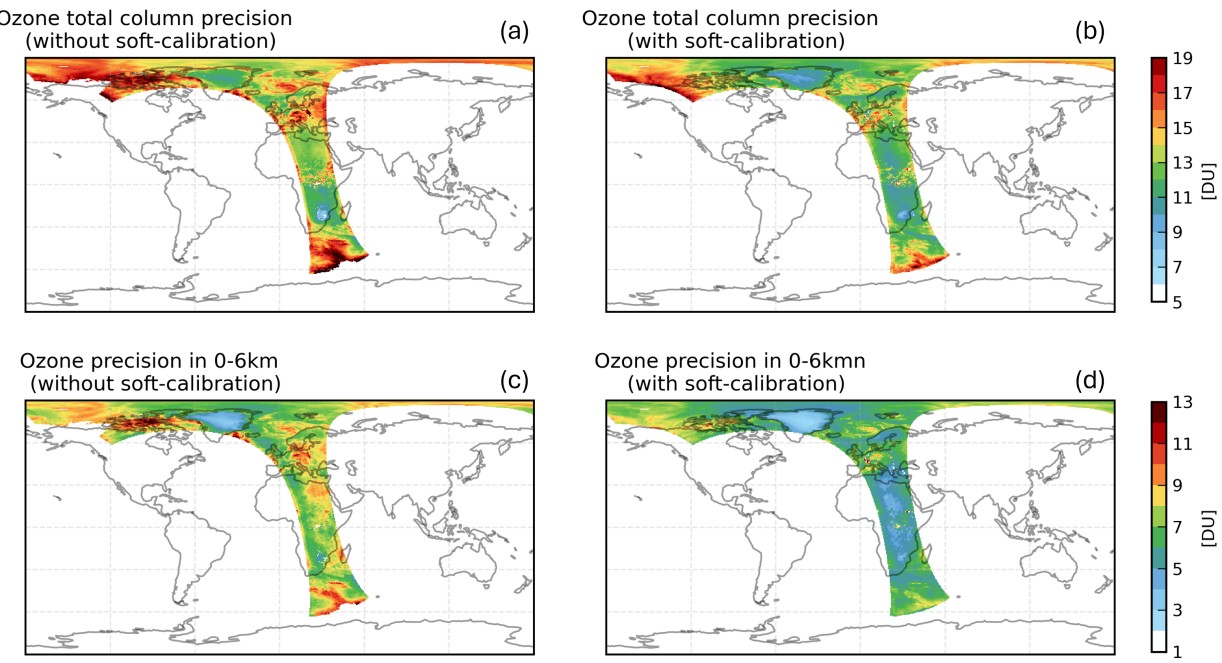

**Figure B1.** Results of orbit number 19452 (on July 15, 2021), regarding: **(a)-(b)** ozone total integrated column precision, **(c)-(d)** ozone 0-6km integrated sub-column precision. Left column shows the retrieval without soft-calibration, right column with soft-calibration.

### B2    Example computation method

Figures (a-c) show the mean absolute radiance residuals computed from the forward model calculations of all the orbits chosen for the year 2021 (see Table A1) as gray dots, while the black and the red (dashed) line represent, respectively, the mean radiance value at the center of the 20 percentile bins and the values obtained from the polynomial fit. The correction (red dashed line) is basically implemented as a piecewise linear correction function of the radiance, wavelength and across-track position, and it can be seen that it has a strong spectral dependence. Figures (d-e) show instead the comparison between the

yearly combined correction (red dashed line) and the single orbit correction of 2021, for the same across-track position and three wavelengths. The seasonal dependence is smoothed out in the combined orbit.



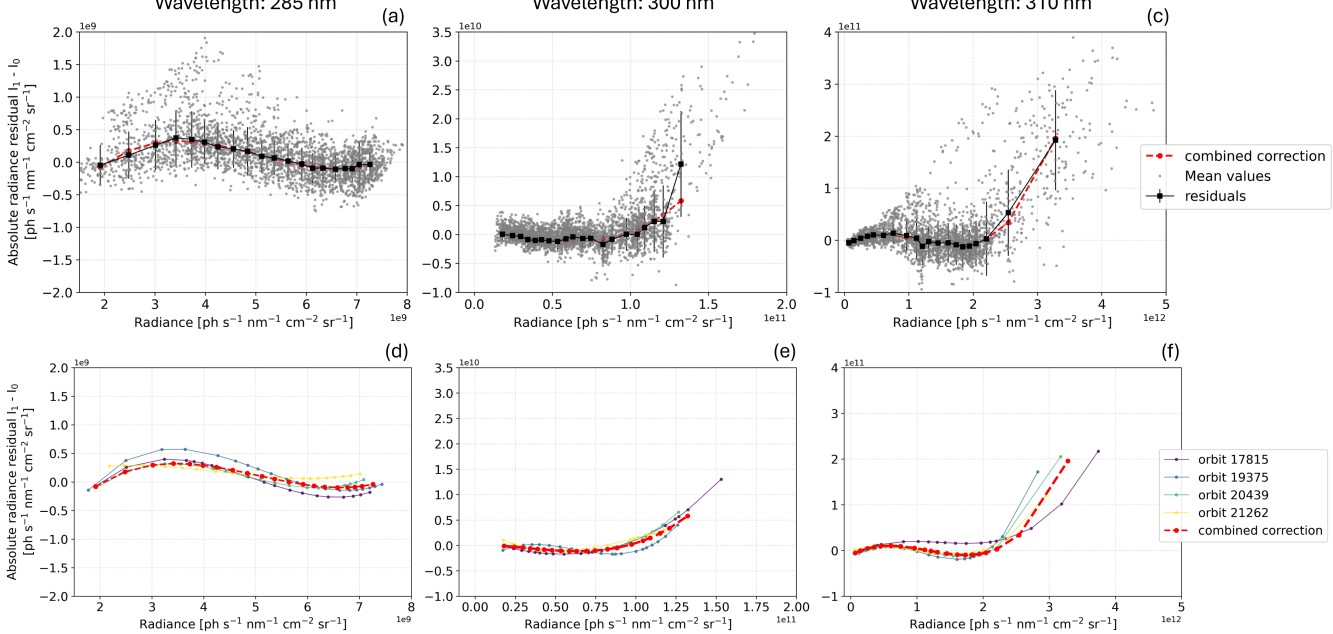

**Figure B2.** Example of the 2021 soft-calibration correction for one across-track position number 35 and three wavelengths. Figures **(a-c)** show the polynomial fit as a red (dashed) line, the mean and standard deviations values as squared markers (black solid line), and the forward model calculations in gray points. Figures **(d-e)** show the combined (polynomial fit) correction in the dashed thick red line, while the single orbit correction in solid thin lines.

## B3  Radiance levels

The soft-calibration correction parameters are computed as a function of the radiance levels, as explained in Sect. 4.2. The following image show an indication of radiance level of the 20 radiance bins for orbit 22992, in band 1 (267–300 nm) and band 450  2 (300–330 nm). Units are converted again in [ph$^{-1}$ cm$^{-2}$ nm$^{-1}$ sr$^{-1}$ s$^{-1}$]





, from the moles of photons.

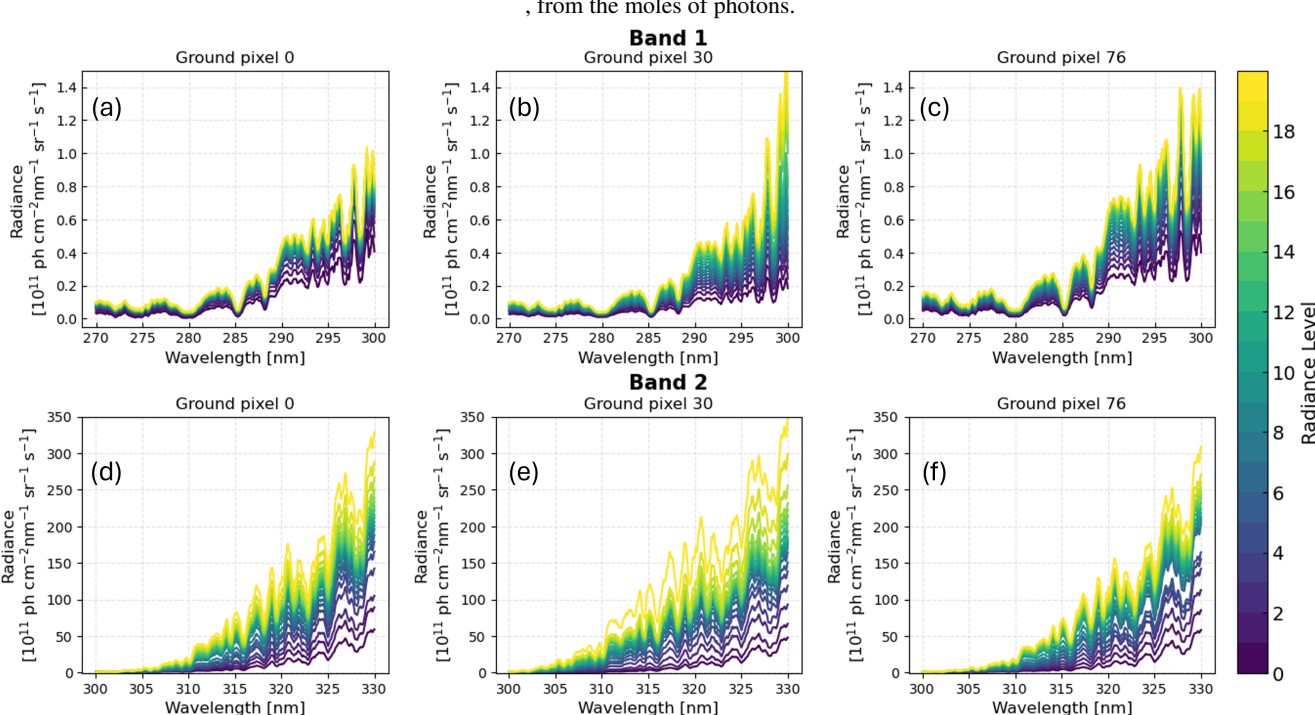

**Figure B3.** Radiance levels of the 20 bins used for the computation of the soft-calibration correction parameters, as explained in Sect. 4.2, for three across-track indices 0, 30, 76. Figures **(a)-(c)** show the radiance levels in band 1 (267–300 nm), while **(d)-(f)** in band 2 (300–330 nm).

## Appendix C: Improvements and results


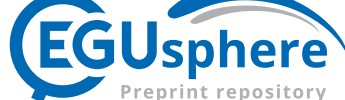

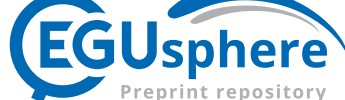

**Figure C1.** The $\sigma$ of the zonal mean of the retrieved quantities shown in Figure 11, using the operational (left column) and the new (central column) soft-calibration correction. The mean and the standard deviation of the $\sigma$ is shown to compare quantitatively the results of the two test orbits. The mean of the standard deviation slightly decrease when using the new soft-calibration.



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
