# Peer review of "Characterization of the UV radiometric calibration for the TROPOMI operational ozone profile retrieval algorithm"

_EGUsphere, 2025_

## Referee Comment (RC1)

**General comments**

Serena Di Pede et al. introduce the upcoming updates to L0–L1 reprocessing within the TROPOMI/S5P framework and describes the impact of adapting the new L1 product into the L2 ozone profile retrieval. The performance of the ozone profile retrieval is highly sensitive to the stability of radiometric and wavelength calibrations. In this context, the soft calibration applied to ozone profile retrieval may serve as a useful diagnostic tool for evaluating the quality of the L1 product and comparing the effects of the updated L0–L1 reprocessing. I believe this paper fits well within the scope of *Atmospheric Measurement Techniques* (AMT) and recommend its publication after addressing the following aspects.

**Major comment**

1. **Introduction**: I think it is unnecessary to present the importance of ozone and the full history of space-based ozone monitoring in this paper, leading to some duplication in the companion paper by Keppens (2024). Instead, it is better to provide an important aspect on the history/current status of L1B and L2 ozone profile product. Like, the validation results and data application results. I believe there are companion papers that already address the importance of version updates related to stray light and background signal corrections and other calibration issues.

2. **Section 2**. Is the L0–L1 reprocessing planned only for the UV1 and UV2 bands? If not, please provide a brief summary of the updates across all spectral bands. If so, it would be helpful to clearly state that the reprocessing applies only to the UVN module.

3. **Section 4**. Are the updates to the version 2.9.0 ozone profile product limited only to the L1B data and its soft calibration? It appears that only three orbits per year are selected to calculate the soft calibration. I believe this sampling may be insufficient, especially after filtering out cloud-affected pixels. Additionally, the ozone fields selected each year could be inconsistent, potentially affecting the robustness of addressing the temporally varying systematic biases.

4. **Figure 2**. Please take a look at https://www.mdpi.com/2072-4292/17/5/779. This paper also indicates the deeper degradation at Fraunhofer lines in UV1 band over

time.

5. **Figure 13**. The OMPS-NP ozone profile product is similar to SBUV-type products, primarily designed for stratospheric ozone retrievals. However, the authors use it as a validation reference for the entire ozone profile, including the troposphere, which may not be appropriate. Additionally, the citation of Kramarova et al. (2017) is incorrect, as that reference pertains to the OMPS Limb Profiler product, not the Nadir Profiler. Moreover, for stratospheric ozone validation, OMPS-LP would be more suitable than OMPS-NP due to its superior vertical resolution.

6. The impact of the L1B updates on the soft calibration is remarkably large (Figure 9), whereas the resulting impact on the ozone product is relatively minor—only a few percent (Figure 11). It implies that the implemented soft calibration works well for addressing the systematic biases existing in both versions of L1B product.

7. To better emphasize the improvements resulting from the L1B reprocessing, I recommend comparing ozone profiles without applying soft calibration. This approach can reveal more substantial enhancements. For example, Bak et al. (2024) demonstrated improved OMI tropospheric ozone distributions using the Collection 4 L1B product without soft calibration, compared to results obtained with the Collection 3 L1B product where soft calibration was applied (see their Figure 8 vs. Figure 12).

    Bak, J., Liu, X., Yang, K., Gonzalez Abad, G., O'Sullivan, E., Chance, K., and Kim, C.-H.: An improved OMI ozone profile research product version 2.0 with collection 4 L1b data and algorithm updates, Atmos. Meas. Tech., 17, 1891–1911, https://doi.org/10.5194/amt-17-1891-2024, 2024.

8. Following previous comment, the impact of applying soft calibration on ozone profile retrievals should be significantly reduced between existing and upcoming versions, which could emphasize the improvements in both L1B and L2 products. Highlighting this reduction would help demonstrate the improvements made in both the L1B and L2 products. In particular, the decreased dependence on soft calibration is an important advancement worth emphasizing.

**Minor comments.**

Line 33: in (Singer et al., 1957) ➔ in Singer et al. (1957)

First line of page 5: from (et al.) ➔ from et al.

I think there are several unnecessary parentheses throughout the manuscript—for example, phrases like "(but on the same detector)", "(and, consequently, corrected for)", and "(and its uncertainty)" could be integrated more smoothly into the main text. Please consider editing these to improve readability and flow.

Figure 9. "residual" -> "background" in caption. Through the manuscript, the background term is used instead of residual.

No availability section is provided in particular for validation reference (OMPS-NP) and the existing operational product, a plan for the upcoming versions.

---

## Author Comment (AC1)

**RC1 - answers**

**General comments**

Serena Di Pede et al. introduce the upcoming updates to L0–L1 reprocessing within the TROPOMI/S5P framework and describes the impact of adapting the new L1 product into the L2 ozone profile retrieval. The performance of the ozone profile retrieval is highly sensitive to the stability of radiometric and wavelength calibrations. In this context, the soft calibration applied to ozone profile retrieval may serve as a useful diagnostic tool for evaluating the quality of the L1 product and comparing the effects of the updated L0–L1 reprocessing. I believe this paper fits well within the scope of Atmospheric Measurement Techniques (AMT) and recommend its publication after addressing the following aspects.

We thank the reviewer for the comments and suggestions. We have addressed each point raised and, where necessary, adjusted the manuscript.

**Major comments**

- 1. Introduction: I think it is unnecessary to present the importance of ozone and the full history of space-based ozone monitoring in this paper, leading to some duplication in the companion paper by Keppens (2024). Instead, it is better to provide an important aspect on the history/current status of L1B and L2 ozone profile product. Like, the validation results and data application results. I believe there are companion papers that already address the importance of version updates related to stray light and background signal corrections and other calibration issues.
  - Thank you for the suggestion and for giving us the chance to improve the manuscript. We have updated the introduction section accordingly. There are more references to processor version updates and companion papers.
- 2. **Section 2:** Is the L0–L1 reprocessing planned only for the UV1 and UV2 bands? If not, please provide a brief summary of the updates across all spectral bands. If so, it would be helpful to clearly state that the reprocessing applies only to the UVN module.
  - The L0-L1 reprocessing updates are limited to the UVN module, no changes are applied to the SWIR detector (bands 7 and 8). The dynamic straylight correction is only applied to bands 1-2, the residual correction is applied to all the bands (from 1-6). In band 3, the so-called "sharp detector feature" is also applied.

    Manuscript change: we clarified this aspect in the Introduction and Section 2.
- 3. **Section 4:** Are the updates to the version 2.9.0 ozone profile product limited only to the L1B data and its soft calibration? It appears that only three orbits per year are selected to calculate the soft calibration. I believe this sampling may be

insufficient, especially after filtering out cloud-affected pixels. Additionally, the ozone fields selected each year could be inconsistent, potentially affecting the robustness of addressing the temporally varying systematic biases.

Thank you for pointing out that this aspect is not clearly explained in the manuscript. Our work describes the updates in the L0-1B processing, and their effect on the soft-calibration correction. As we mention at the beginning of Section 4 (Improvements, in the updated version of the manuscript), there was no change in the soft-calibration procedure itself. The soft-calibration correction is computed per each year of the mission considering 5 orbits per year, always over the Pacific Ocean where the total ozone variability is low and to ensure consistency during the years. The orbits used for the forward model comparisons are always chosen in the same months of the year, specifically in January, March, July, September and November, to capture seasonality.

Manuscript change: We moved the text section in 272-279 to the procedure (Method section 3.3) to enhance clarity on the methodology.

4. **Figure 2:** Please take a look at https://www.mdpi.com/2072-4292/17/5/779. This paper also indicates the deeper degradation at Fraunhofer lines in UV1 band over time.

Thank you for the suggestion. Figure 2 of this paper shows in a clear way the degradation of the UV1 band over time, while the stability of the other bands.

Manuscript change: we added this work as a reference of the manuscript.

5. **Figure 13:** The OMPS-NP ozone profile product is similar to SBUV-type products, primarily designed for stratospheric ozone retrievals. However, the authors use it as a validation reference for the entire ozone profile, including the troposphere, which may not be appropriate. Additionally, the citation of Kramarova et al. (2017) is incorrect, as that reference pertains to the OMPS Limb Profiler product, not the Nadir Profiler. Moreover, for stratospheric ozone validation, OMPS-LP would be more suitable than OMPS-NP due to its superior vertical resolution.

The initial idea was to compare the ozone profile retrieved from two nadir-viewing instruments. However, we decided to replace Figure 13 with a new figure, motivated in the answer to question 7. We believe that the new figure is more consistent with the scope and analysis of the manuscript. The inter-satellite comparison will be discussed in another manuscript, when a larger dataset over the mission will be used to enable a more appropriate comparison of the two data versions.

Manuscript change: replace Figure 13, as motivated in question 7.

6. The impact of the L1B updates on the soft calibration is remarkably large (Figure 9), whereas the resulting impact on the ozone product is relatively minor—only a few percent (Figure 11). It implies that the implemented soft calibration works well for addressing the systematic biases existing in both versions of L1B product.

Yes, that is correct. The soft-calibration procedure is not changed and it should address well the systematic biases shown by both L1B versions. We would like to remark that Figure 11 shows the results of the retrieval of a single orbit (over the Pacific Ocean) which was also used to test the calibration results with different L1B versions (shown in Figure 9). We believe that the largest impact on the ozone retrieval will be visible when looking at the comparison of the two ozone product versions of an extended dataset over the whole mission. This comparison would enhance more subtle differences regarding, for example, the bias of the data or the drift compared to on-ground measurements.

Manuscript change: Building on this remark and connecting to questions 7-8, we replace Figure 11-12-13 using along-track averages and the across-track dependent anomalies metric to show the results on the ozone profile, as these metrics are more sensitive to the L1B updates presented in the manuscript. Moreover, the figures are based on a retrieval of a full day (15 July 2024) instead of a single orbit (22992).

7. To better emphasize the improvements resulting from the L1B reprocessing, I recommend comparing ozone profiles without applying soft calibration. This approach can reveal more substantial enhancements. For example, Bak et al. (2024) demonstrated improved OMI tropospheric ozone distributions using the Collection 4 L1B product without soft calibration, compared to results obtained with the Collection 3 L1B product where soft calibration was applied (see their Figure 8 vs. Figure 12). Reference: Bak, J., Liu, X., Yang, K., Gonzalez Abad, G., O'Sullivan, E., Chance, K., and Kim, C.-H.: An improved OMI ozone profile research product version 2.0 with collection 4 L1b data and algorithm updates, Atmos. Meas. Tech., 17, 1891–1911, https://doi.org/10.5194/amt-17-1891-2024, 2024.

Thank you for the suggestion and the opportunity to improve the analysis. We agree that the comparison of ozone profile retrievals without applying the soft calibration correction can be a useful diagnostic tool to reveal more subtle differences. We looked into this comparison using the ozone retrieval of a full day (15 July 2024) and replaced the figures 12-13.

**Manuscript change:**

- Figure 11 has been replaced by the retrieval results as a function of the acrosstrack position (along-track averages). We believe that this metric is more consistent with the analysis of the manuscript as it enhances the effect of the across-track dependent calibration biases;
- Figures 12-13 have been replaced by the global maps and the across-track dependent anomalies of the total and tropospheric ozone column (the 0-6km ozone sub-column is in the Appendix E1). These figures also display the

comparison with the two versions of the retrieval but without applying the softcalibration correction.

The text has been accordingly updated.

8. Following previous comment, the impact of applying soft calibration on ozone profile retrievals should be significantly reduced between existing and upcoming versions, which could emphasize the improvements in both L1B and L2 products. Highlighting this reduction would help demonstrate the improvements made in both the L1B and L2 products. In particular, the decreased dependence on soft calibration is an important advancement worth emphasizing.

We agree that the impact of applying the soft-calibration correction on the retrieval should be reduced when using the updated version of the L1B data. If we indeed look at the difference between the retrieval, with and without soft-calibration (but same data version), we notice that the impact of the soft-calibration is reduced when using the updated version. This can be seen in the following global maps, showing the difference between the retrieved total and tropospheric ozone, with and without soft-calibration, for the same data version: version 2.8.0 (L1B 2.0.1) on the left, while the updated version on the right.

Manuscript change: the global maps in Figures 12-13 also show that the impact of the soft-calibration on the retrieval using the updated L1B data is smaller than in the previous version. However, we remark that it is not possible to perform a good quality retrieval without the soft-calibration correction.

**Minor comments**

- Line 33: in (Singer et al., 1957) → in Singer et al. (1957)
   Corrected
- First line of page 5: from (et al.) → from et al.
   Corrected
- 3. I think there are several unnecessary parentheses throughout the manuscript—for example, phrases like "(but on the same detector)", "(and, consequently, corrected for)", and "(and its uncertainty)" could be integrated more smoothly into the main text. Please consider editing these to improve readability and flow. We agree with the reviewer, and we updated the text accordingly to avoid too many parentheses.
- 4. Figure 9. "residual" -> "background" in caption. Through the manuscript, the background term is used instead of residual.
  Thank you for the comment. We updated the manuscript accordingly to use the term "residual".
- 5. No availability section is provided in particular for validation reference (OMPS-NP) and the existing operational product, a plan for the upcoming versions.

  After the implementation of the new L1 3.0.0 and L2\_O3\_PR (probably 2.9.1) data version, there will be more data available for validation, which we plan to address in another publication. We added this information in the Conclusions.

---

## Author Comment (AC2)

**RC2 - answers**

**General comments**

In general, I find this manuscript version improved over the initial submission. The authors have done a better job of organizing results to make clear when corrections are and are not applied. More explicit statements for each figure would still be appreciated. It is clear that the new instrument characterization described in this paper is improving the accuracy of the radiance data (thereby reducing the magnitude of soft calibration corrections), but some of the details regarding how those improvements were achieved is lacking. It is only by providing such details that readers can evaluate the quality of the characterizations and learn something about instrument performance.

We thank the reviewer for the comments and appreciation. We addressed each comment, updating the manuscript where necessary.

**Section 2.2, Lines 135-138**

This text describes the contents of Figure 2, which presents a compelling case that the Mg line signals have increased over time relative to background signals. Unfortunately, the reader has no idea from this figure what the time scales are, since the paper provides no assignment of dates for S5P orbit numbers. Two common reasons these Fraunhofer line changes occur is solar activity and improper correction for additive signal errors. The most likely additive errors are detector dark current and spectral stray light. All three phenomena are valid explanations for why the Mg line depths shown in Fig. 2 are decreasing. The authors state that the Fig. 2 results are independent of solar activity effects, but offer no details supporting this statement. Was a correction applied, and if so what was it? They do not discuss their background signal corrections, which must also increase as the instrument ages. These are glaring omissions if they expect the reader to accept the most surprising conclusion, that internally-scattered stray light is increasing within TropOMI. There are few plausible mechanisms for postlaunch increases in spectrometer scattering (primarily caused by the grating), so the authors need a convincing argument why the least likely of the 3 explanations is in fact the actual cause of the observed signal changes. One convincing metric would be to compare the Mg II core-to-wing ratios from solar irradiance and Earth radiance spectra at different times in the mission. In the absence of additive errors these two ratios should remain in lock step with each other. If, however, spectral stray light is increasing, the effect on Earth radiances should be much greater than on solar irradiances.

The authors highly appreciate the reviewer's detailed analysis. We did not want to exclude that additive effects are present. At line 141 (previous manuscript), we therefore remarked that the additive effects will be discussed later in the manuscript, in the Sections about the straylight and residual correction updates. We agree that all three

phenomena can cause the Fraunhofer line changes (solar activity, uncorrected additive straylight, uncorrected additive dark signal), however it is difficult to disentangle them. Our claim that straylight is one (and not "the") actual cause is based on the following: 1) in-flight growth of signal in the straylight rows; 2) the large changes in the first E2 years, when solar activity is still low (before 2020); 3) the smooth behavior of far-field straylight, showing a single peak at 280nm in comparison with solar activity changes showing a double peak; 4) the improved correlation between Mg and Ca lines time series when implementing the dynamic straylight correction (image included in Appendix A1(b)).

The effect of detector dark current signal in irradiance is small, but it is true that it can be relatively more important for radiance. The residual correction, described in Section 2.3, is implemented in both radiance and irradiance measurements and it is meant to correct for remaining additive effects (therefore also detector dark current and RTS). The soft-calibration spectra, pictured in Figure 9c-d, also show improvements regarding across-track biases when using input data with residual correction implemented.

Regarding the suggestion of comparing the Mg line indices of radiance and irradiance: only if all additive errors - especially in radiance - have been successfully removed, then the correlation of the time series of the line indices are perfectly aligned. However, we do not claim that that is the case, otherwise the bottom panel in Figure 9 would be much smoother.

We indeed reprocessed ~200 orbits over the entire mission that includes both radiance and irradiance. Due to restrictions, it was not possible to regenerate a consistent set of the (ir)-radiances without the mentioned three improvements: the radiance measurements of the existing dataset have been processed with CKD that have been updated several times, with large sensitivities at the 280nm FL. The part of the timeseries that is consistent, from April 2024 onward, indeed shows better correlation (not shown). Manuscript adjustment: Figure 2 has been modified by adding the dates and showing the effects of the dynamic straylight on the peaks at 280nm and 286nm. The text of Section 2.2 has been also updated to add more explanations on the reasoning behind the exclusion of solar activity, with companion plots in Appendix A1 and A2, showing improved correlation between the Mg and Ca line indices for irradiance measurements. The correlation improves, however it is not perfect.

**Section 2.2.1, Line 153**

The statement that the SL correction algorithm is a 2D correction that only addresses inband stray light is confusing. I suspect the authors mean the correction address both spatial and spectral stray light, but the latter is limited to only photons from within the same band. If so, a clarification of this point would be helpful.

The kernel is 2D, and its convolution with the 2D (spatial-spectral) detector image results in a straylight image containing both the spatial and spectral straylight. The kernel operates on an entire detector image, e.g. UV, VIS, NIR or SWIR detectors, with each

detector consisting of two electronic bands. Perhaps here the term "band" is causing confusion.

Manuscript adjustment: we deleted the phrase in line 154 as the out-of-spectral range straylight is not relevant for the manuscript and it interrupts the flow regarding the inband straylight. We updated also the text of this section to enhance clarity.

**Section 2.2.1, Lines 179-189**

The authors should provide a clearer explanation of how the SL convolution kernels are adjusted. What criteria were used to assess the correct spectral and spatial distributions? Scattering occurring after the entrance slit typically has no preferential spatial or spectral direction. The 2D scattering PSF is usually symmetric. Therefore, the large difference in the two orthogonal cross sections of that PSF (shown in Fig. 3) is rather surprising and the authors are encouraged to provide more detail about why they feel the shapes should behave so differently. I presume that telescope stray light (i.e. pre-slit scattering) has been excluded from the plots in Fig. 3a. Please state so explicitly.

Yes, symmetric scattering has indeed been our first assumption. The problem is that the on-ground calibration measurements (white light source measurements with small spatial range) led to non-unique solutions for the convolution kernel. Giving the onground measurements, non-axisymmetric kernels were in principle equally valid as solutions.

The first E2 radiance measurements were clearly over-corrected in band 1, resulting in negative radiance signals after the straylight correction. Given that radiance images have more variation in the spectral than spatial direction, the revised kernel shape, stronger in the spectral dimension, would be less strong in the spectral direction while keeping all its other properties deduced from the on-ground calibration measurements. This is why the elliptical kernel shape was considered an improvement over the original symmetrical shape: it is still a valid (alternative) solution allowed within the entire set of on-ground measurements, and giving better corrections on the in-flight measurements.

Manuscript adjustment: We modified section 2.2.1 to clarify the choice of the elliptical kernel shape. A new image has been added in Appendix to show the comparison among the original, the elliptical kernel shape, and a third case with a long tails kernel shape. All these kernel shapes are valid given the on-ground calibration measurements.

**Section 2.4**

The authors mention the introduction of time-varying SL correction to account for the increasing Mg II signals and the SL row signals, but they provide no details about how this is implemented. Are the kernel shapes altered in time? If so, how? Are the spatial and spectral components of the kernel locked relative to each other or allowed to vary independently? Whatever the approach, it will be helpful if the authors can demonstrate

the effect of these dynamic corrections on the metrics shown in Figures 2 and 4. Figure 10 does indicate the effect on RTM residuals of these instrument corrections, but it does not address the effect on irradiance or stray light rows.

Thanks for the comment and the opportunity to improve the clarity of the text and the analysis. The convolution kernel shape (both the original and elliptical version) is static in time. The dynamic straylight correction is instead based on the instantaneous measurements obtained from the straylight rows, hence the attribute "dynamic". The only difference between the original and elliptical kernel is the asymmetry in the spectral dimension, as the mass (sum of all elements) of the convolution kernel is the same in both cases. Again, we mention that we make use of some freedom in proposing slightly different kernels that are still valid solutions with respect to the on-ground measurements. To show the effect of the introduction of the dynamic correction on the degradation of un-corrected irradiance measurements, we updated Figure 2.

Regarding Figure 10: the soft-calibration is applied on radiance measurements.

Manuscript adjustment: Figure 2 has been updated with the following new figure, showing the effect of the dynamic straylight correction on the degradation of uncorrected irradiance measurements. Section 2.2.1 was also updated to clarify the difference between the updated convolution kernel (static in time) and the newly added dynamic (time dependent) straylight correction.

**Section 4.3**

Regarding the observed residuals (as represented by soft calibration corrections) presented in Figures 7 & 8 the authors do not discuss whether or not a solar activity correction has been applied. The increase seen in Figure 7a is consistent with the activity increase over the mission timeline. Figure 8 also appears to be consistent with the increase. An accurate solar activity correction must be applied prior to residual analysis, therefore the authors should provide details about any correction that was applied.

What data was it derived from? What is the magnitude of the changes in time at key band 1 wavelengths?

There is no solar activity correction applied to the radiance and irradiance spectra used for the comparison with forward model calculations.

Section 3.1 describes the correction steps applied to the L1 (bands 1-2) measurements before the soft-calibration correction (see Fig. 5), as pre-processing steps of the ozone profile retrieval: spectral calibration, spatial regridding, spectral binning, signal-to-noise ratio floor, polarization & raman correction.

We add the following image to show the magnitude of the absolute difference between the un-corrected radiance and the modeled radiance at key band 1 wavelengths and at two band 2 wavelengths, for a single central ground pixel number 35. The solid line ("oper") stands for the current processing, while the dashed line ("new") shows the time change of the absolute residuals using the updated bands 1-2 measurements. These figures also clearly show the decreased magnitude of the bias and the decrease time dependence of the bias over the mission.

Manuscript adjustment: we realized it might not have been clear that the bands 1-2 preprocessing described in Section 3.1 were the corrections applied to the input spectra before the soft-calibration correction. Therefore, we moved the description of the measurements pre-processing under the main section on the *TROPOMI UV soft calibration*. Moreover, we changed the last step in Figure 5 from "Radiometric calibration correction" to "Soft-calibration correction" to enhance clarity.